# Testing electron–phonon coupling for the superconductivity in kagome metal CsV$_3$Sb$_5$

Yigui Zhong [1,10], Shaozhi Li[2,10], Hongxiong Liu[3,10], Yuyang Dong[1], Kohei Aido[1], Yosuke Arai[1], Haoxiang Li[2,4], Weilu Zhang[1,5], Youguo Shi[3], Ziqiang Wang[6], Shik Shin [1,7], H. N. Lee [2], H. Miao [2] ✉, Takeshi Kondo [1,8] ✉ & Kozo Okazaki [1,8,9] ✉

In crystalline materials, electron-phonon coupling (EPC) is a ubiquitous many-body interaction that drives conventional Bardeen-Cooper-Schrieffer super-conductivity. Recently, in a new kagome metal CsV$_3$Sb$_5$, superconductivity that possibly intertwines with time-reversal and spatial symmetry-breaking orders is observed. Density functional theory calculations predicted weak EPC strength, λ, supporting an unconventional pairing mechanism in CsV$_3$Sb$_5$. However, experimental determination of λ is still missing, hindering a micro-scopic understanding of the intertwined ground state of CsV$_3$Sb$_5$. Here, using 7-eV laser-based angle-resolved photoemission spectroscopy and Eliashberg function analysis, we determine an intermediate λ=0.45–0.6 at $T$ = 6 K for both Sb 5$p$ and V 3$d$ electronic bands, which can support a conventional super-conducting transition temperature on the same magnitude of experimental value in CsV$_3$Sb$_5$. Remarkably, the EPC on the V 3$d$-band enhances to λ~0.75 as the superconducting transition temperature elevated to 4.4 K in Cs(V$_{0.93}$Nb$_{0.07}$)$_3$Sb$_5$. Our results provide an important clue to understand the pairing mechanism in the kagome superconductor CsV$_3$Sb$_5$.

The kagome lattice, made of corner-shared triangles, is an exciting platform for emergent quantum phenomena[1–3]. Due to the wave-function interference, the electronic structure of the kagome lattice features flat band, Dirac fermion, and van Hove singularities that result in a rich interplay between topology, geometry, and correlations[4,5]. For kagome metals near the van Hove singularities, the high density of states combining with the frustrated lattice geometry are predicted to support novel electronic orders[6–8]. Recently, in a topological kagome metal CsV$_3$Sb$_5$, superconductivity that possibly intertwines with charge density wave (CDW)[9–12] (Fig. 1a), nematicity[13–16] and loop current[17,18] is observed. To date, the origin of superconductivity and its interplay with the other symmetry-breaking orders remain rigorous debate. Angle-resolved photo-emission spectroscopy (ARPES) studies[19,20] observed multiple van Hove singularities from V 3$d$-electrons near the Fermi level ($E_F$), highlighting electronic driven instabilities[6,8] (Fig. 1b). Furthermore, the density functional theory (DFT) calculated EPC strength[21], λ~0.25, in CsV$_3$Sb$_5$ fails to support the superconducting transition temperature[9], $T_c$~2.6 K, indicating unconventional pairing mechan-ism. However, a recent ARPES study of a cousin compound KV$_3$Sb$_5$ revealed a clear kink[22] in the electronic band structure near the van Hove singularity, suggesting a moderate EPC. Therefore, an

[1]Institute for Solid State Physics, The University of Tokyo, Kashiwa, Chiba 277-8581, Japan. [2]Material Science and Technology Division, Oak Ridge National Laboratory, Oak Ridge, TN 37831, USA. [3]Beijing National Laboratory for Condensed Matter Physics and Institute of Physics, Chinese Academy of Sciences, 100190 Beijing, China. [4]Advanced Materials Thrust, The Hong Kong University of Science and Technology (Guangzhou), 511453 Guangzhou, Guangdong, China. [5]Department of Engineering and Applied Sciences, Sophia University, Tokyo 102-8554, Japan. [6]Department of Physics, Boston College, Chestnut Hill, MA 02467, USA. [7]Office of University Professor, The University of Tokyo, Kashiwa, Chiba 277-8581, Japan. [8]Trans-scale Quantum Science Institute, The University of Tokyo, Bunkyo, Tokyo 113-0033, Japan. [9]Material Innovation Research Center, The University of Tokyo, Kashiwa, Chiba 277-8561, Japan. [10]These authors contributed equally: Yigui Zhong, Shaozhi Li, Hongxiong Liu. ✉e-mail: miaoh@ornl.gov; kondo1215@issp.u-tokyo.ac.jp; okazaki@issp.u-tokyo.ac.jp

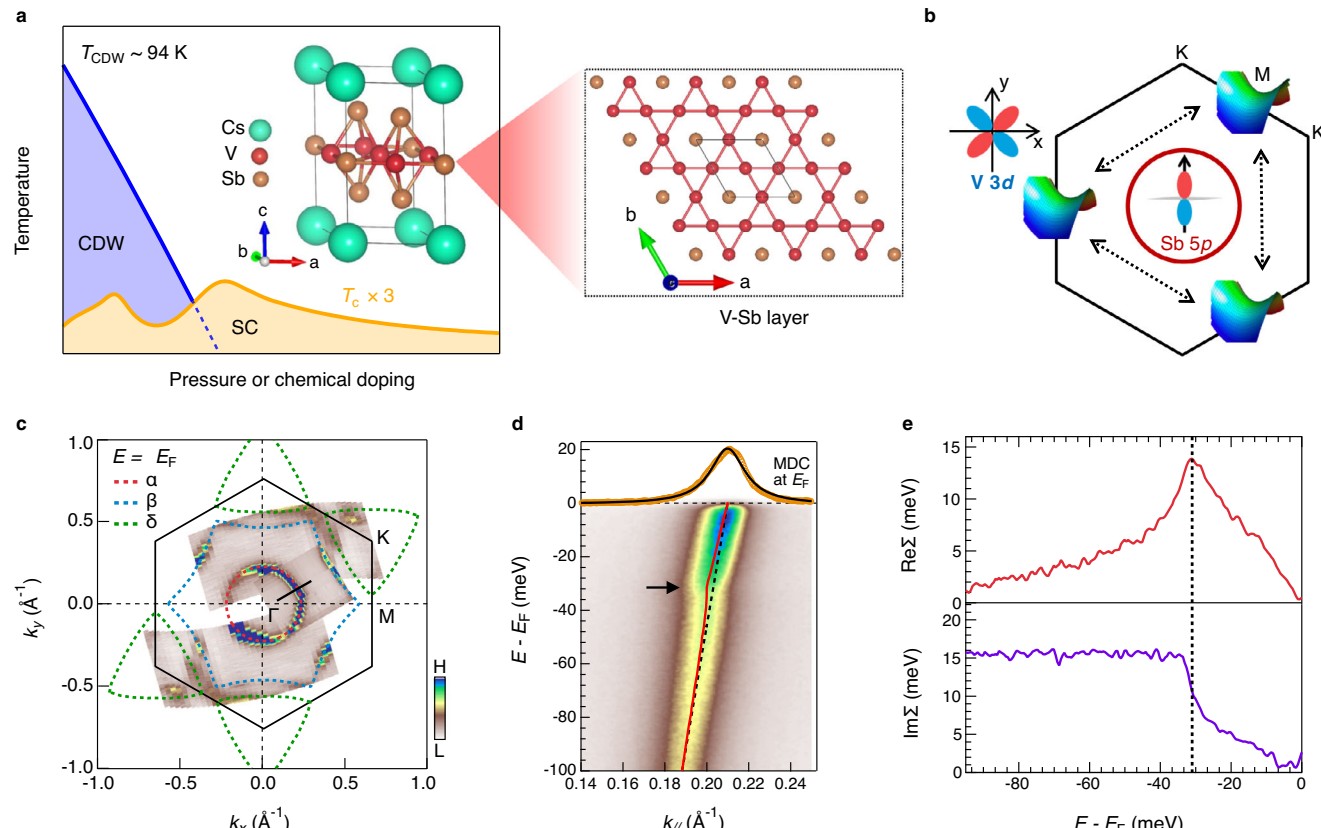

**Fig. 1 | EPC-induced electronic kink in CsV₃Sb₅.** **a** Schematic temperature versus pressure/doping phase diagram. The inset shows the crystal structure of $CsV_3Sb_5$. A top view of the V-Sb layer is zoomed in on the right panel. **b** Schematic of van Hove singularities at the M point of the Brillouin zone boundary which are connected by three nesting wavevectors. The von Hove singularities has mainly V 3$d$ orbital characters ($d_{x^2-y^2}$ and $d_{yz}$). The circular electronic pocket at Γ point has mainly Sb 5$p_z$ orbital character. **c** FS mapping with intensity integrated within $E_F \pm 5$ meV. The FS sheet in $k_y < 0$ is symmetrized from the one in $k_y > 0$ and a superposed FS sheet is collected from another independent sample. Dashed lines are DFT determined FSs. **d** ARPES intensity plot, corresponding to the black cut in **c**, showing a kink in the band dispersion. The band dispersion extracted from the MDCs is overlaid as a red curve. The dashed black line represents the bare band. The arrow indicates the position of the kink. The MDC at $E_F$ and its Lorentzian fit are shown as yellow and black lines, respectively. **e** Real-part self-energy ReΣ($\omega=E$-$E_F$) and imaginary-part self-energy ImΣ($\omega=E$-$E_F$). A background of ImΣ$_{other}$ is subtracted for ImΣ($\omega$) (see supplementary note 2). The dashed black line marks the energy position of the kink.

experimental estimation of orbital- and momentum-dependent λ and its possible connection with superconductivity are highly desired to understand the nature of the superconductivity in $CsV_3Sb_5$. Here we experimentally extract the orbital- and momentum-dependent $λ_{p,d}(\mathbf{k})$ by determining the EPC-induced kinks in the electronic band structure. Our results reveal an intermediate EPC with $λ$=0.45–0.6 in $CsV_3Sb_5$, which can support a $T_c$ on the same magnitude of the experimental value. Intriguingly, we find that $λ_d$ is enhanced by about 50% in the isovalent-substituted $Cs(V_{0.93}Nb_{0.07})_3Sb_5$ with an elevated $T_c = 4.4$ K. Our results suggest that EPC can play an important role on the superconductivity in $CsV_3Sb_5$.

## Results

Figure 1a, c shows the crystal structure and Fermi surface (FS) topology of $CsV_3Sb_5$, respectively. In agreement with previous DFT and ARPES studies[12,19–21], the Sb 5$p$-band forms a circular FS, marked as α, at the BZ center and the V 3$d$ bands yield hexagonal and triangle FSs, marked as β and δ in Fig. 1c, respectively. Figure 1d shows a typical ARPES intensity plot of the α band corresponding to the black cut shown in Fig. 1c. The coupling between electrons and bosonic modes is manifested by the intensity and dispersion anomalies, known as kink[23,24], near a binding energy $E_B$~32 meV. This many-body effect can be quantified by fitting the ARPES momentum distribution curves (MDCs)

with a Lorentzian function[25]:

$$I(\mathbf{k}, \omega) \propto A(\mathbf{k}, \omega) = \frac{1}{\pi} \frac{\text{Im}\Sigma(\omega)}{(\omega - \varepsilon(\mathbf{k}) - \text{Re}\Sigma(\omega))^2 + \text{Im}\Sigma(\omega)^2}, \quad (1)$$

where ReΣ($\omega = E$-$E_F$) and ImΣ($\omega = E$-$E_F$) are the real and imaginary parts of the single-particle self-energy. ε($\mathbf{k}$) is the non-interacting bare band that can be approximated as a liner dispersion crossing $E_F$[23]. Figure 1e demonstrates the extracted self-energy of the α band. We subtract a linear bare band from the experimentally extracted band to obtain ReΣ($\omega$) (see supplementary note 1). To extract the electron–boson coupling induced ImΣ($\omega$), the electron-electron and electron-impurity scatterings induced self-energy effects are removed, as suggested by previous practices[23,26] (see supplementary note 2). At $E_B$~32 meV, a peak near in ReΣ($\omega$) and a step jump in ImΣ($\omega$) prove strong many-body interactions. Since the self-energy anomalies persist above CDW transition temperature $T_{CDW}$ (supplementary Fig. S7), we attribute the self-energy anomaly to EPC.

Figure 2a–c compares the EPC-induced kinks on the α and β bands at 6 K. The ARPES intensity plots of the α and β bands shown in Fig. 2b correspond to the black cuts in Fig. 2a. While the kink near $E_B$~32 meV is clear on both the α and β bands, an additional kink is observed at a lower $E_B$~12 meV on the β band (Fig. 2c). The 12-meV kink is also

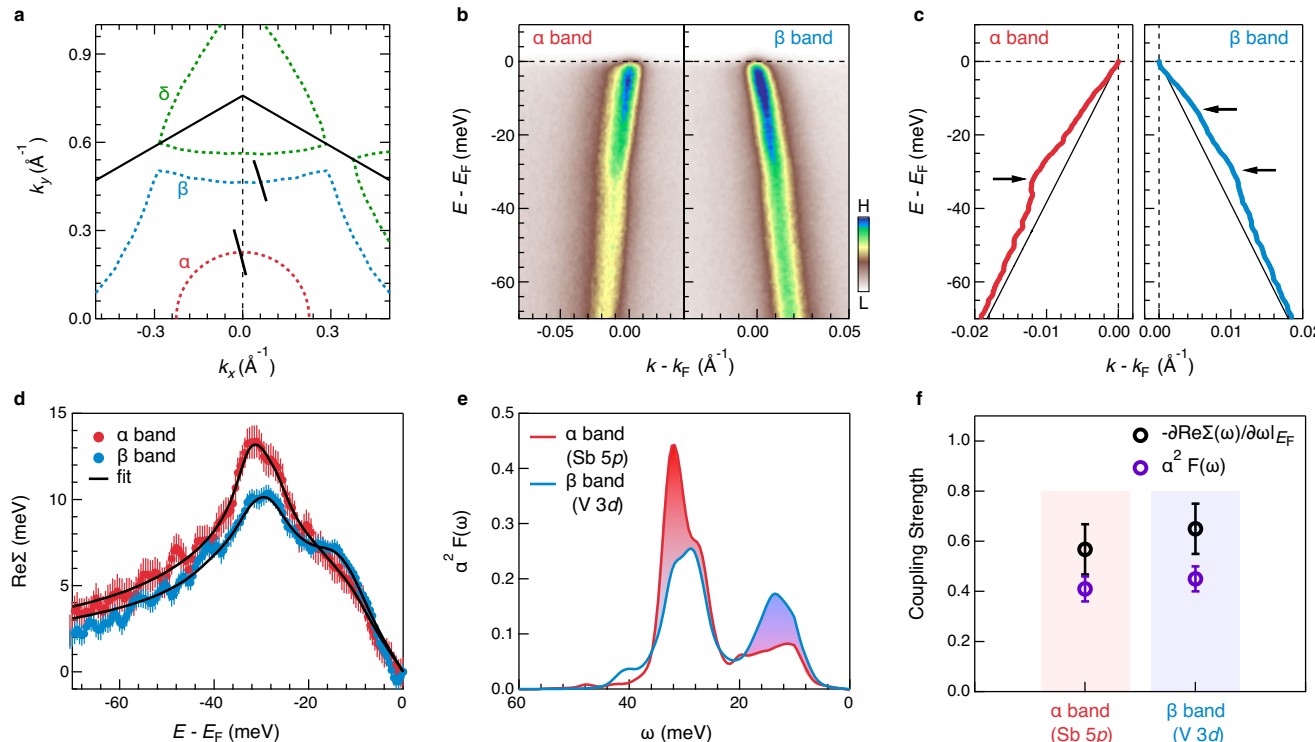

**Fig. 2 | Orbital dependent EPC. a** Contours of the FSs and the momentum location of the cuts shown in (**b**). **b** ARPES intensity plots the α and β bands. The momentum is rescaled with respect to their $k_F$. **c** Extracted band dispersions of the α and β bands. The arrows show the energy position of the kinks. The black lines are the corresponding bare bands. **d** Extracted ReΣ(ω) of the α and β bands. The error bars for ReΣ(ω) are determined from standard deviation of the MDC fits, which is converted to energy by multiplying velocity of bare band. The black lines are the ReΣ(ω) reproduced by maximum entropy method. **e** Extracted Eliashberg coupling functions $α^2F(ω)$ for the α and β bands. **f** $λ_{p,d}$ estimated from $α^2F(ω)$ (purple circles) and $λ_{dev}$ defined by the slope of ReΣ(ω) at $E_F$ (black circles). The error bars are determined by the standard deviation of the ReΣ(ω).

prominent in ReΣ(ω). As we show in Fig. 2d, ReΣ(ω) of the β band shows a peak near $E_B$-12 meV, proving strong $d$-electron–phonon coupling near 12 meV. In contrast, ReΣ(ω) of the α band only shows a broad shoulder.

The observation of clear EPC effects on both $5p$ and $3d$ bands points to a non-neglectable role of EPC for superconductivity in $CsV_3Sb_5$. To test the EPC-driven superconductivity, we extract the Eliashberg function, $α^2F(ω)$, at $T = 6$ K, slightly above $T_c$, using the maximum entropy method[27,28] (see methods). A fit of the ReΣ(ω) and the extracted $α^2F(ω)$ are shown in Fig. 2d, e, respectively. λ and the logarithmic mean phonon frequency are obtained via[28,29]:

$$λ = 2\int_0^{ω_{max}} \left[\frac{α^2F(ω)}{ω}\right]dω, \qquad (2)$$

$$\ln ω_{log} = 2/λ \int_0^{\infty} \ln ω \left[\frac{α^2F(ω)}{ω}\right]dω, \qquad (3)$$

where $ω_{max}$ is the maximum frequency of the phonon spectrum. As shown in Fig. 2e, the orbital dependence of the EPC is mirrored in the different shapes of $α^2F(ω)$, where phonon modes near 32 meV are accounted for 70% of the total EPC strength on the α band, $λ_p$, but less than 50% for the EPC strength on the β band, $λ_d$. Interestingly, due to the spectral weight redistribution in $α^2F(ω)$ (shaded area in Fig. 2e), the extracted $λ_p$ and $λ_d$ are similar with $λ_{p,d}$-0.45 ± 0.05. We also employed the MEM fits the extracted ImΣ(ω), which yields a λ consistent with the ReΣ(ω) fits (see supplementary note 4 and Fig. S2). Theoretically, λ can approximately be derived from a simpler approach[29] following $λ_{dev} = -∂ReΣ(ω)/∂ω|_{ω=E_F} \cong λ$, when $T$ is far lower than the Debye temperature. At $T = 6$ K, this method yields a $λ_{dev}$-0.6±0.1, qualitatively consistent with Eq. (2) within the experimental uncertainty (Fig. 3f).

Generally, EPC can exhibit momentum dependence. Figure 3 summarizes the momentum-dependent kinks on the α and β bands. The ARPES intensity plots and extracted band dispersions along representative directions are shown in Fig. 3a, b and Fig. 3d, e, respectively (see supplementary Fig. S3 for complete dataset). The extracted $λ_{dev}(\boldsymbol{k})$ for the α and β bands of two independent samples (supplementary Fig. S4) are summarized in Fig. 3f, which shows a nearly isotropic behavior within experimental uncertainties.

The orbital- and momentum-dependent results demonstrate that the EPC strength λ in $CsV_3Sb_5$ falls in the intermediate range of 0.45–0.6, which is about 2 times larger than the previous DFT predicted $λ_{DFT}$-0.25 (ref. 21). Using McMillan's formula[30] and taking the lower and upper limits of the experimentally estimated λ and the logarithmic mean phonon frequency -17.1 meV obtained from Eq. (3), we derive $T_c$ in a range from 0.8 K to 3 K (see supplementary note 6). The upper limit is comparable to the experimentally determined $T_c$ in $CsV_3Sb_5$ (Fig. 4a). We shall note that the CDW gap near the M point[20,31] (supplementary Fig. S6) flattens the δ bands near $E_F$, hindering the precise estimation of EPC strength. However, strong self-energy anomalies are observed on the δ bands and they have the same energy scales as the α and β bands (supplementary Fig. S6).

As shown in Figs. 4a and 1a, $T_c$ of $CsV_3Sb_5$ is increased with chemical substitutions or external pressure[32–35]. We thus continue to examine the EPC in a 7% Nb-doped $Cs(V_{0.93}Nb_{0.07})_3Sb_5$ with $T_c$-4.4 K[34]. Electronic Kinks are observed on both the α and β bands as shown in Fig. 4b for the ARPES intensity plots and Fig. 4c for the extracted band dispersions. Figure 4d shows the extracted ReΣ(ω) on the α and β bands of $Cs(V_{0.93}Nb_{0.07})_3Sb_5$. The shaded area corresponds to the

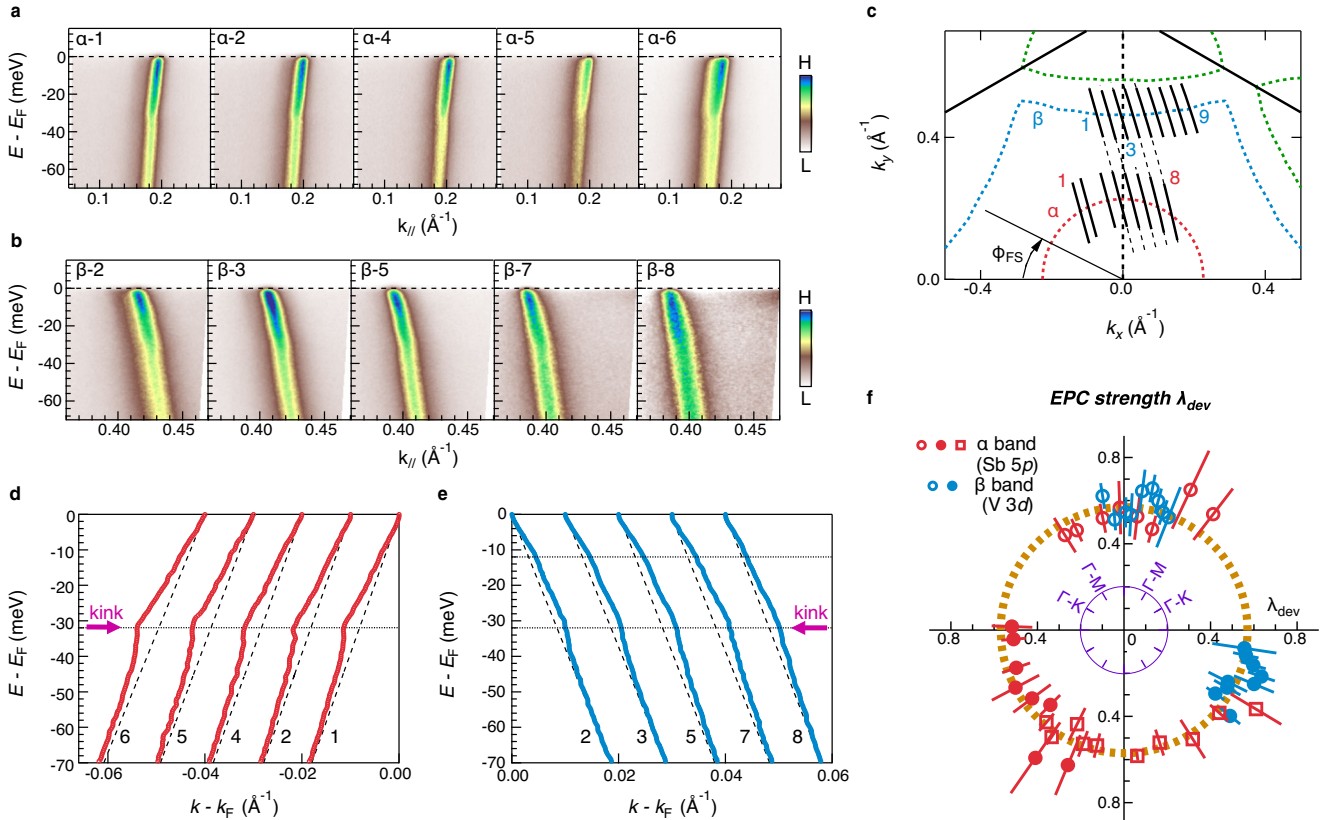

**Fig. 3 | Momentum dependence of the electronic kink. a, b** Representative ARPES intensity plots for the α and β bands as marked in **c**, respectively. **d, e** Extracted band dispersions for the representative α and β bands, respectively. The dashed black lines are bare bands. The red purple arrows indicate the 32-meV kink. These extracted dispersions are offset horizontally for a better view. **f** EPC strength $\lambda_{dev}$ defined by the slope of ReΣ(ω) at $E_F$ plotted with FS angle $\phi_{FS}$ as the hollow markers. The solid markers are mirrored from the hollow makers for a better clarity. The square markers are the data collected from another independent sample (supplementary Fig. S4). The error bar for the $\lambda_{dev}$ is determined by the standard deviation of the ReΣ(ω). The dashed yellow line represents the average value of $\lambda_{dev}$. The definition of the FS angle $\phi_{FS}$ is shown in **c**.

ReΣ(ω) of the pristine $CsV_3Sb_5$. Remarkably, we observe that while ReΣ(ω) on the α band is similar in $Cs(V_{0.93}Nb_{0.07})_3Sb_5$ and $CsV_3Sb_5$, on the β band, it shows a strong enhancement in the Nb-doped sample, especially near $E_B$-10 meV. Based on the extracted $α^2F(ω)$, shown in Fig. 4e, we find that $λ_d$-0.75 ± 0.05 is enhanced by about 50% in $Cs(V_{0.93}Nb_{0.07})_3Sb_5$ (Fig. 4f). Such giant enhancement is also manifested by the slope of ReΣ(ω) near $E_F$ (Fig. 4d). Consequently, the enhanced $λ_d$ in $Cs(V_{0.93}Nb_{0.07})_3Sb_5$ is expected to elevate $T_c$ up to 4.5 K (see supplementary note 6), which is comparable to the experimental value of 4.4 K (Fig. 4a). Such synchronous enhancements of $λ_d$ and $T_c$ may indicate that the V 3*d*-electron–phonon couplings are the main driver of the superconductivity in $CsV_3Sb_5$.

Finally, we discuss the influences of CDW order on the quantitative extraction of λ at $T < T_{CDW}$. The formation of a CDW gap will modify the bare band to deviate from a linear dispersion near $E_F$. As we show in the supplementary Fig. S5, within the experimental resolution, we do not observe a CDW gap on the α and β bands. Therefore, for the α and β bands, the CDW modified bare band dispersion below $T_{CDW}$ is

$\sqrt{\varepsilon_0^2(k)+\Delta_{CDW}^2} \cong \varepsilon_0(k)$, where $\varepsilon_0(k)=v_0\hbar k$ is the linear bare band dispersion above $T_{CDW}$. In this case, the linear bare band assumption used in our study is a good approximation. Indeed, the excellent agreement of ReΣ(ω) and ImΣ(ω) linked by Kramers-Kronig transformation[23,26] validates the linear bare band assumption for the α and β bands (supplementary Figs. S1c-d). The linear bare band assumption, however, does not apply to the δ band that forms a CDW gap comparable to the kink energy[20,31]. We also note that the formation of CDW will also modify the electronic self-energy. As we show in the

supplementary Fig. S7e, $λ_{dev}$ shows an inflection point at $T_{CDW}$, which may suggest an enhanced EPC strength below $T_{CDW}$. However, it can also be a consequence of the CDW-corrected electronic self-energy effect (see supplementary note 8).

In summary, by investigating the electronic kinks, we determined an intermediate EPC that is twice larger than the DFT calculated value in the kagome superconductor $CsV_3Sb_5$ and $Cs(V_{0.93}Nb_{0.07})_3Sb_5$. Our results provide an important clue to understand the pairing mechanism in $CsV_3Sb_5$. The orbital, momentum of electronic kinks and their strengthening with the promoted $T_c$ prove that the EPC in $CsV_3Sb_5$ is strong enough to support a $T_c$ comparable to the experiment value and hence cannot be excluded as a possible pairing mechanism. While the exact microscopic pairing mechanism calls for further scrutiny, it is important to point out that the EPC-driven superconductivity is not incompatible with the recently observed pair-density wave (PDW) in $CsV_3Sb_5$[17]. Indeed, PDW has been observed in another conventional superconductor $NbSe_2$, where the pair-density modulation is due to the real space charge density modulations[36]. We also note that the EPC-driven superconductivity can coexist with the time-reversal symmetry-breaking (TRSB) orders or fluctuations[18,37,38], as proposed by theoretical studies[39–41]. In those cases, the superconducting order parameter is expected to intertwine with the TRSB order parameter, which gives rise to an unconventional ground state.

## Methods
### Growth and characterization of single crystals
Single crystals of $CsV_3Sb_5$ were grown using $CsSb_2$ alloy and Sb as flux. Cs, V, Sb elements and $CsSb_2$ precursor were sealed in a Ta crucible in a

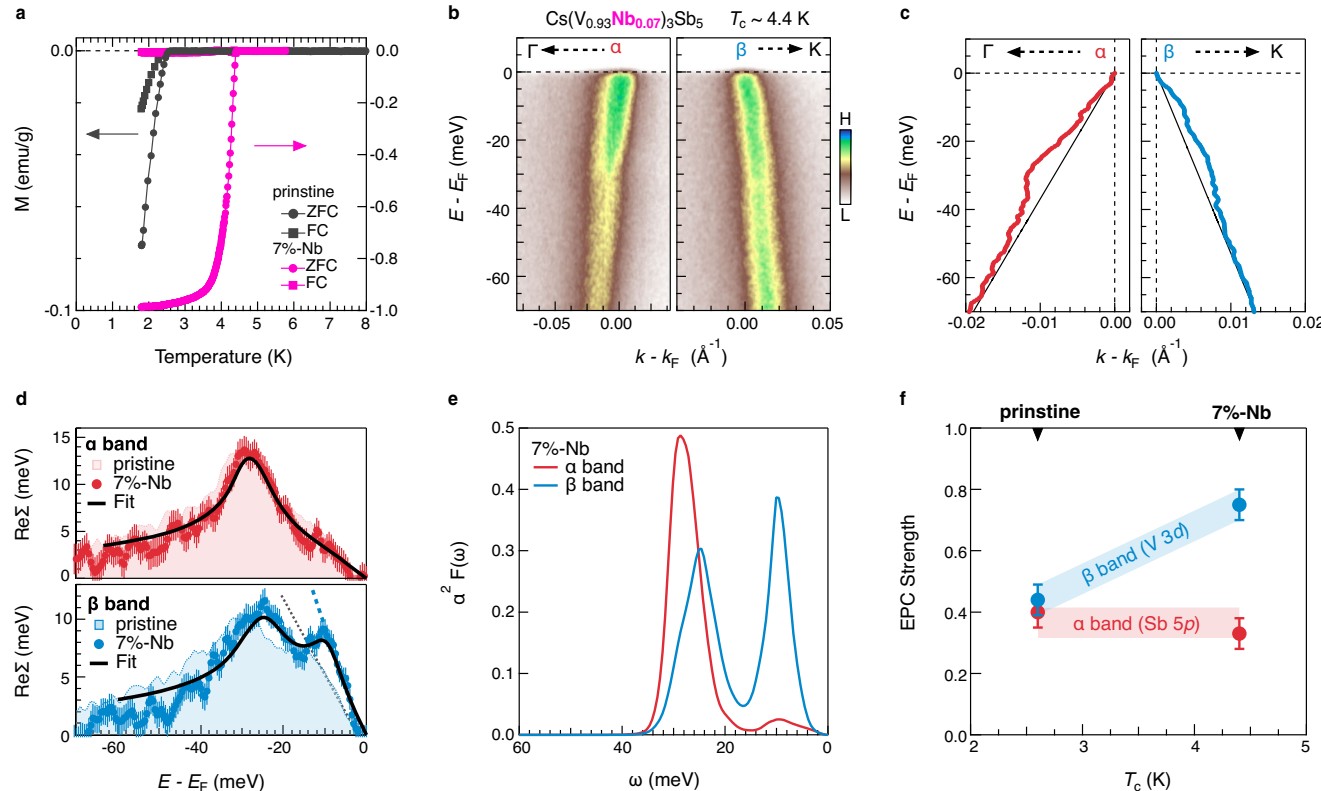

**Fig. 4 | Orbital and energy selective enhancement of EPC in Cs(V₀.₉₃Nb₀.₀₇)₃Sb₅.**
**a** Temperature dependence of magnetic susceptibility for pristine CsV₃Sb₅ and
Cs(V₀.₉₃Nb₀.₀₇)₃Sb₅. Both zero-field cooling (ZFC) and field cooling (FC) curves are
presented. **b** ARPES intensity plots of the α and β bands along the Γ−K direction for
Cs(V₀.₉₃Nb₀.₀₇)₃Sb₅. The false color is adjusted for better visualization. **c** Extracted
band dispersions for the α and β bands shown in (**b**). The black lines are the
corresponding bare bands. **d** Extracted ReΣ(ω) for the α (top panel) and β (bottom
panel) bands. The error bars for ReΣ(ω) are determined from the standard

deviation of the MDC fits, which is converted to energy by multiplying the velocity
of the bare band. The ReΣ(ω) of pristine CsV₃Sb₅ is plotted as colored shadows for a
direct comparison. The solid black curves are the reproduced ReΣ(ω) via maximum
entropy method. The dashed blue and gray line represents the slope of the ReΣ(ω)
at $E_F$ in Cs(V₀.₉₃Nb₀.₀₇)₃Sb₅ and CsV₃Sb₅, respectively. **e** Extracted Eliashberg
function α²F(ω). **f** EPC strength λ estimated from the α²F(ω), which is plotted as a
function of $T_c$. The error bar for λ is determined by the standard deviation of
the ReΣ(ω).

molar ratio of 1:3:14:10, which was finally sealed in a highly evacuated
quartz tube. The tube was heated up to 1273 K, maintained for
20 hours and then cooled down to 763 K slowly. Single crystals were
separated from the flux by centrifuging. The single crystals of
Cs(V₀.₉₃Nb₀.₀₇)₃Sb₅ were provided by Jinggong New Materials
(Yangzhong) Co., Ltd. The growth and characterizations of
Cs(V₀.₉₃Nb₀.₀₇)₃Sb₅ were present in ref. 34. The magnetic susceptibility
was measured under magnetic field 20 Oe for CsV₃Sb₅ and 5 Oe for
Cs(V₀.₉₃Nb₀.₀₇)₃Sb₅.

### Laser-ARPES measurements
ARPES measurements were performed for the freshly cleaving surface
with a Scienta-Omicron R4000 hemispherical analyzer with an ultra-
violet laser (hν = 6.994 eV) at the Institute for Solid State Physics, the
University of Tokyo[42]. The energy resolution was set to be 1.3 meV. The
sample temperature was set to be 6 K if there is no special
announcement. The samples were cleaved in situ and kept under a
vacuum better than 3 × 10⁻¹¹ torr during the experiments.

### Maximum entropy method
The Eliashberg function $\alpha^2F(\omega;\epsilon,\boldsymbol{k})$ is related to the real part of the
self-energy by the integration function

$$\mathrm{Re}\Sigma(\epsilon,\boldsymbol{k};T) = \int_0^\infty \mathrm{d}\omega\, \alpha^2F(\omega;\epsilon,\boldsymbol{k})K\left(\frac{\epsilon}{k_BT},\frac{\omega}{k_BT}\right), \qquad (4)$$

where $K(y,y') = \int_{-\infty}^{\infty}\mathrm{d}x\frac{f(x-y)2y'}{x^2-y'^2}$ and $f(x)$ is the Fermi distribution
function. It is an ill-posed problem to obtain the Eliashberg function
from Eq. (4). In this work, we adopted the maximum entropy method
(MEM)[27,28], which is frequently used to perform the analytic
continuation[43]. By considering the energy resolution of the laser-
ARPES, we estimated that the error bar of the real part of the self-
energy was 1 meV. MEM requires a model default function to define the
entropic prior. Here, we adopted the following model:

$$m(\omega) = \begin{cases} m_0\left(\frac{\omega}{\omega_D}\right)^2 & \omega \le \omega_D \\ m_0 & \omega_D \le \omega \le \omega_m \\ 0 & \omega > \omega_m \end{cases}, \qquad (5)$$

where $m_0 = 15$ meV, $\omega_D = 10$ meV, and $\omega_m = 80$ meV. This default model
was also used in the previous study of the electron–phonon coupling
on the Be surface[28].

### Data availability
Data are available from the corresponding author upon reasonable
request.

### Code availability
Codes are available from the corresponding author upon reasonable
request.

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

## Acknowledgements

The authors thank Yaoming Dai, Kun Jiang and Binghai Yan for stimulating discussions, and thank Yongkai Li, Jinjin Liu and Zhiwei Wang for technical supports. This research was sponsored by the U.S. Department of Energy, Office of Science, Basic Energy Sciences, Materials Sciences and Engineering Division (x-ray scattering and theoretical analysis), and by Grants-in-Aid for Scientific Research (KAKENHI) (Grant Nos. JP18K13498, JP19H01818, JP19H00651 and JP21H04439) from the Japan Society for the Promotion of Science (JSPS), by JSPS KAKENHI on Innovative Areas "Quantum Liquid Crystals" (Grant No. JP19H05826), by

the Center of Innovation Program from the Japan Science and Technology Agency, JST, and by MEXT Quantum Leap Flagship Program (MEXT Q-LEAP) (Grant No. JPMXS0118068681), and by MEXT as "Program for Promoting Researches on the Supercomputer Fugaku" (Basic Science for Emergence and Functionality in Quantum Matter Innovative Strongly-Correlated Electron Science by Integration of "Fugaku" and Frontier Experiments, JPMXP1020200104) (Project ID: hp200132/hp210163/hp220166). Y.G. S is supported by the National Natural Science Foundation of China (Grants No. U2032204), and the Strategic Priority Research Program of the Chinese Academy of Sciences (Grants No. XDB33030000). Z.Q.W is supported by the U.S. Department of Energy, Basic Energy Sciences Grant No. DE-FG02-99ER45747.

## Author contributions

Y.Z. and H.M. conceived the project. Y.Z. performed the ARPES experiment with the assistant from Y.D., K.A., and Y.A. and the guidance from T.K. and K.O.; T.K., K.O., and S.S. constructed the 7-eV laser-based ARPES system. Y.Z. S.L., Z.W., W.Z. and H.M. performed the theoretical analysis and simulations. H.X.L. and Y.G.S. grown the samples. H.X.L., H.N.L., and H.M. performed structural characterizations. Y.Z., H.M., T.K., and K.O. prepared the manuscript with inputs from all authors.

## Competing interests

The authors declare no competing interests.
