## [Peer Review File · Nature Communications]

Testing Electron-phonon Coupling for the Superconductivity in Kagome Metal CsV₃Sb₅Reviewers' Comments:

Reviewer #1:

Remarks to the Author:

In the manuscript, Zhong and co-workers worked on characterizing electron-phonon coupling (EPC) and its possible role on superconducting mechanism. Using high-resolution angle-resolved photoemission spectroscopy (ARPES) with 7 eV laser, they mapped Fermi surfaces (FSs) and band dispersions from V and Sb bands. They first characterized the real part of self-energies on each alpha, beta, and delta bands, and afterward put careful analysis efforts for alpha and beta bands which originates from Sb 5p and V 3d orbitals, respectively. While the alpha band has a single kink structure at 32 meV, the beta band has double kink structure at 12 and 32 meV. Eliashberg function and EPC strength λ are obtained via maximum entropy method. The λ was double-checked with slope of the real part of the self-energy afterward. The inferred EPC strength is larger than that from density-functional-theory calculations results, thus they claimed that EPC coupling can now phenomenologically explain superconducting transition temperatures. Finally, they remarked that the kink structures are isotropic.

The ARPES data quality is high and the demand for knowledge on EPC in AV3Sb5 kagome metals is huge among the community. Thus, I would expect that this work can attract notable interests. Meanwhile, I also note that comprehensiveness of the data is bit thin and there should be additional experimental supports to provide consistency of their data interpretation. Those are my prerequisites and some points in order to consider this manuscript for publication in Nature Communications.

1) I would like to emphasize that ARPES is one of the most powerful experimental techniques to determine single-particle self-energy analysis. Despite its advantages, one needs to pay careful attention to spectra as extrinsic effects may join and dilute out intrinsic natures. Therefore, it is desirable to prove consistency for the self-energy analysis. There can be many ways, and one of them I can suggest is utilizing the Kramers-Kronig (K-K) relation. It seems like that quality of the experimental dataset must be good enough to infer imaginary part of the self-energy as well as the real part. Especially, observation on the double-kink structure from the beta band needs such consistency check.

2) Additional to the comment 1), I also need to elaborate that AV3Sb5 kagome metals have finite k_z -dispersion though it is often underrated. Note that self-energy analysis, especially line-width analysis to obtain imaginary part of the self-energy, can make sense after confirming that k_z broadening does not affect (or minimally affect) linewidths in momentum-distribution curves. Thus, I ask the authors to check k_z dispersion from their own data or published data whether it is critical to perform K-K analysis. I am already aware of that presented experimental data with 7 eV laser can be ideal as it locates in a "good" position in the universal curve to minimize k_z broadening effect. Despite this advantage, the elaborated point should be quantitatively checked prior to K-K analysis.

3) Is it more reasonable to use DFT-calculated band dispersions for "bare bands" as Luo et al. (reference 22 in the main text) did? There should be enough published results that potentially work fine as "bare bands" in CsV3Sb5.

4) The authors claimed that the alpha band has single-kink structure while the beta band has double-kink. In their supplementary material, Fig. S1 show momentum dependence of the kink structures, and I see that the 12 meV kink structure in the beta band becomes unclear away from the G-K symmetry line. Can the authors back up this data or explain why?

5) The momentum distribution curve at EF shown in Fig. 1d does not look like a single peak. Was the sample free from superposition with any misaligned domain? I am aware of that CsV3Sb5 crystals cleave well but can leave small pieces/domain behind.

6) Note that evaluation for λ using the slope of the self-energy may be inappropriate as temperature increases so it becomes non-negligible compared to the Debye temperature (E.W. Plummer et al., Prog. Surf. Sci. 74, 251 (2003).).

Reviewer #2:
Remarks to the Author:
See the attached file.

The manuscript by Zhong et al. presents measurements of the kinks in electronic structure of a recently established Kagome superconductor CsV₃Sb₅ by laser ARPES. By the Eliashberg function analysis, they determine the electron-phonon coupling (EPC) strength for two bands with different orbital characters. This value is suggested to be comparable to the T_c of CsV₃Sb₅ and the superconductivity is suggested to be driven by the EPC. Overall, the authors present high quality experimental data on an interesting material, however, it is a stretch to directly draw such a strong conclusion that the superconductivity originates from the EPC based on current results, because the connection between the EPC strength and the underlying superconducting pairing glue is not straightforward. Extensive works have been recently done on the superconductivity of CsV₃Sb₅ by various techniques, but few of them are referred and discussed here on the basis of current results. Rather than the brief arguments in the manuscript, the implications of the experimental data and the connections with previous studies are suggested to be further explored and discussed. Moreover, the analysis is not sufficient to unambiguously determine the value of EPC strength with only the real part of the self-energy being considered, it is suggested to further include the analysis via the relation between the imaginary part of the self-energy and the Eliashberg function. Hence, I don't recommend the publication in Nature Communications and suggest transferring the manuscript to a more specialized journal. Below I make some further comments.

1. In Fig. 1e, the correspondence is not clear by showing the FWHM of MDCs together with the real part of the self-energy. It is suggested to present the imaginary part of the self-energy instead of FWHM.
2. In Line 74-76 (Fig. 2d), an additional kink is observed around 12 meV, which is prominent in the beta band and less prominent in the alpha band, what is the cause of this orbital dependent behavior? Can we obtain some illumination from this for the superconductivity?
3. The argument on the phonon hardening effect is confusing. The self-energy anomaly of the beta band around 12 meV remains almost the same upon warming (Fig. S4d), while the neutron studies (Ref. 26) show no evident change of the phonon energy above T_{cdw}. Why the 3d-EPC only hardens the phonons when T < T_{cdw}?
4. In Fig. S4, the kink around 30 meV in the beta band becomes much less prominent at higher temperature, in contrast to that of the alpha band, what is the cause of the different temperature dependence?
5. The Eliashberg function analysis in the current manuscript is not sufficient to obtain an unambiguous EPC strength because only the real part of the self-energy is

examined, which depends strongly on the choice of the bare band, quite different λ values could be obtained by a small change of the bare band. While not doubting the current results, the authors are suggested to further examine the EPC strength by the Eliashberg function analysis with calculating both the real part and imaginary part of the self-energy.

6. In Line 108-109, “We shall note that while the delta band also displays strong self-energy anomaly near the same energy scale”, this argument is confusing. Although two kink-like features are observed in the delta band in Fig. S3, the one close to EF could be mainly affected or induced by the CDW gap opening of the delta band instead of EPC, as illustrated in Fig. S3b, S3c, and S3f, where the larger CDW gap size the flatter delta band near EF. It is suggested to discuss the contribution of CDW gap opening in this kink-like structure.
7. As implied by the arguments in Line 110-111 (main text) and Line 63-64 (supplementary), CDW seems to compete with superconductivity for the available density of states near EF. Furthermore, as shown in Fig. S3f, the CDW gap of the delta band shows an evident momentum dependence as in Ref. 22. Therefore, one would expect an anisotropic superconducting gap opening of the delta band due to the competition between CDW and superconductivity. This could contradict the current conclusion drawn in the manuscript.
8. Related to the above point, although the EPC strength can support a T_c comparable to the experimental value, here the connection between the EPC strength and the underlying superconducting pairing glue in CsV₃Sb₅ is not straightforward. In order to draw such a strong conclusion, the authors are suggested to measure the superconducting gap symmetry in the momentum space. Otherwise, I suggest the authors could soften this point in the manuscript.
9. Some typos in the supplementary. Line 60, Fig. S3d → Fig. S3f; Fig. S3a, gamma → delta; Line 80, liens → lines.

Reviewer #3:

Remarks to the Author:

In this laser-based ARPES work, Zhong and coworkers extract the momentum dependence of the electron-phonon coupling constant λ in CsV₃Sb₅ Kagome metal via analysis of the electron-phonon renormalization of the electronic band dispersion (kink analysis). The authors extract a λ in the 0.45-0.6 range, a factor two larger than what is predicted by DFT, supporting conventional electron-phonon-mediated superconductivity in CsV₃Sb₅.

Before expressing my opinion about this work's novelty and potential impact, the authors should thoroughly answer the following technical questions/comments.

1) The authors extract λ at low temperature. However, based on Fig. S4 where the temperature dependence of $\text{Re}\Sigma$ is presented for both alpha and beta bands, the kink strength decreases when T increases (suppression of around 20-40% for both alpha and beta bands – the weakening of the kink is easily observable by superimposing $\text{Re}\Sigma$ curves at different T). Given that (i) there is no clear dependence of $\text{Re}\Sigma$ on the CDW transition, and (ii) it is unlikely that the EPC is changing over T , the authors should extract λ at high temperature where the CDW is absent. Indeed, any gapping of the electronic states (even if localized in a few areas of the momentum space) may lead to a positive feedback effect enhancing the kink strength [for instance, see Li et al. Nature Communications 9, 26 (2018)].

The only way to extract a reliable λ is to analyze the kink for $T > T_{\text{CDW}}$. However, I am afraid that for $T > T_{\text{CDW}}$ the extracted λ will be smaller than what is reported in the current manuscript (and possibly matching with DFT calculations), thus invalidating the authors' conclusions.

2) The authors extract the Eliashberg function and related λ using the maximum entropy method (MEM), as shown in Fig. 2. In this regard, what bare dispersion was used? In addition to the $\text{Re}\Sigma$ of Fig. 2d, the authors should display the $\text{Im}\Sigma$ extracted via MEM and compare it to the experimental $\text{Im}\Sigma$.

Moreover, the authors estimate the momentum dependence of λ by computing the first derivative of $\text{Re}\Sigma$ at EF. This method is unreliable for quantitative extraction of λ (it is blind to other possible underlying contributions, such as e-e). Indeed, the λ extracted by MEM is approximately 50% lower than the one retrieved by the first derivative of $\text{Re}\Sigma$, clearly showing the unreliability of the first-derivative method. The authors should apply the MEM at high temperature to extract the momentum-dependence of λ .

Reviewer #1:

Thank you for your careful review of our manuscript. Below we copied your report *in black* and describe how we have edited the manuscript in response to your comments *in blue*.

Comment 1: The ARPES data quality is high and the demand for knowledge on EPC in AV₃Sb₅ kagome metals is huge among the community. Thus, I would expect that this work can attract notable interests. Meanwhile, I also note that comprehensiveness of the data is bit thin and there should be additional experimental supports to provide consistency of their data interpretation. Those are my prerequisites and some points in order to consider this manuscript for publication in Nature Communications.

Reply 1: We thank Reviewer #1's high evaluation of our work, especially on its potential impact to the community. Following Reviewer #1's suggestions, we made the following improvements in the revised manuscript:

- (1) We added new experimental data of an isovalent-substituted Cs(V_{0.93}Nb_{0.07})₃Sb₅ with $T_c \sim 4.4$ K. Our self-energy analysis shows that the EPC strength is enhanced by about 50% on the V $3d$ band (Fig. 4 of the revised manuscript). Based on the extracted Eliashberg function, we obtained an enhanced T_c close to the experimental value. This new result strengthens the main conclusion of this work that EPC plays a key role in the emergence of superconductivity in CsV₃Sb₅.
- (2) We extracted the Eliashberg function using real- and imaginary-part self-energy and checked their consistency using the Kramers-Kronig relation.

Comment 2: 1) I would like to emphasize that ARPES is one of the most powerful experimental techniques to determine single-particle self-energy analysis. Despite its advantages, one needs to pay careful attention to spectra as extrinsic effects may join and dilute out intrinsic natures. Therefore, it is desirable to prove consistency for the self-energy analysis. There can be many ways, and one of them I can suggest is utilizing the Kramers-Kronig (K-K) relation. It seems like that quality of the experimental dataset must be good enough to infer imaginary part of the self-energy as well as the real part. Especially, observation on the double-kink structure from the beta band needs such consistency check.

Reply 2: Thanks for the valuable suggestion. Following Reviewer #1's suggestion, we conducted Kramers-Kronig (K-K) transformation and confirmed the consistency between the real- and imaginary-part self-energy. Figure R1.1 shows the imaginary (top) and real (bottom) parts of the self-energy extracted from our ARPES data. The black curve on top of the experimental real-part self-energy is obtained by using K-K transformation of the imaginary-part self-energy. The good consistency validates the bare band assumption used in our manuscript. *We added this comparison in the revised supplementary materials (supplementary note 1 and Fig. S1).*

We further compared the Eliashberg functions extracted from the real- and imaginary-part self-energy in Fig. S2 of the supplementary materials. Finally, we employed a self-consistent fit on the experimentally extracted band dispersions starting from a trial bare velocity for the α and β band in CsV₃Sb₅. The converged bare-band velocity validates our bare band assumption (*see supplementary note 2 and Fig. S2 in the revised manuscript*).

Figure R1.1. Consistency-check of the self-energy following Kramers-Kronig relation. **a**, Imaginary part of the self-energy $\text{Im}\Sigma$ (top panel) and real part of the self-energy $\text{Re}\Sigma$ (bottom panel) for the α band. **b**, Same with **a** but for the β band. The black lines in the bottom panel are the real part transformed from $\text{Im}\Sigma$ following the Kramers-Kronig relation. A constant background is subtracted for $\text{Im}\Sigma$. The α and β band here are the ones presented in Fig. 2 of the main text.

Comment 3: Additional to the comment 1), I also need to elaborate that in AV3Sb5 kagome metals have finite k_z -dispersion though it is often underrated. Note that self-energy analysis, especially line-width analysis to obtain imaginary part of the self-energy, can make sense after confirming that k_z broadening does not affect (or minimally affect) linewidths in momentum-distribution curves. Thus, I ask the authors to check k_z dispersion from their own data or published data whether it is critical to perform K-K analysis. I am already aware of that presented experimental data with 7 eV laser can be ideal as it locates in a “good” position in the universal curve to minimize k_z broadening effect. Despite this advantage, the elaborated point should be quantitatively checked prior to K-K analysis.

Reply 3: This is an excellent point. As Reviewer #1 correctly pointed out, the k_z broadening extrinsically broadens the linewidth, and hence can potentially affect the extracted self-energy. This broadening depends on details of the band dispersion along k_z as well as the electron coherence along the surface normal. DFT calculations predicted a neglectable k_z -dependence of the β band near E_F [Ref. 21, *Physical Review Letters* **127**, 046401 (2021)]. For the α band, as shown in Fig. R1.2 [*Physical Review Research* **4**, 033072(2022)], its k_z -dependence is also neglectable within ~ 200 meV below E_F . Since our linewidth analysis from the ARPES spectra was performed within 100 meV below E_F , the k_z broadening is considered to have a minimal effect. Indeed, the K-K transformation shown in Fig. R1.1 also supports our estimation. This is because the extrinsic k_z broadening is not expected to obey the K-K relation.

To elaborate on this point, we added the above discussion in the supplementary materials (supplementary note 1).

Figure R1.2 The k_z dependence of the α band in CsV_3Sb_5 . The figure is adopted from *Physical Review Research* **4**, 033072(2022).

Comment 4: 3) Is it more reasonable to use DFT-calculated band dispersions for “bare bands” as Luo et al. (reference 22 in the main text) did? There should be enough published results that potentially work fine as “bare bands” in CsV_3Sb_5 .

Reply 4: We considered using the DFT-calculated band dispersion as the “bare band”. However, there are some challenges to directly use them as “bare bands” in real practice. According to the present and published ARPES data, there are finite renormalization effects compared with the DFT-calculated band dispersions due to the weak but finite electron-electron correlations. In addition, the chemical potential determined by ARPES measurements is different from the DFT calculations. To overcome these challenges, within a narrow energy range (~ 100 meV), we use a linear dispersion as the bare band which is commonly assumed to analyze the kink dispersion of the high- T_c cuprates superconductors [*Phys. Rev. Lett.*, **87**, 177007 (2001); *Phys. Rev. Lett.* **110**, 217006 (2013)], iron-based superconductors [*Phys. Rev. B* **83**, 134513 (2011)], transition metal dichloride charge density wave materials [*Phys. Rev. Lett.* **92**, 086401 (2004)], and other materials in the ARPES community. As shown in Fig. R1.1, the extracted real and imaginary parts of the self-energy based on the assumed linear bare band consistently satisfy the Kramers-Kronig relation, validating our bare band assumption.

Comment 5: 4) The authors claimed that the alpha band has single-kink structure while the beta band has double-kink. In their supplementary material, Fig. S1 show momentum dependence of the kink structures, and I see that the 12 meV kink structure in the beta band becomes unclear away from the G-K symmetry line. Can the authors back up this data or explain why?

Reply 5: Reviewer #1 made a very good point. First, we would like to clarify that the 12 meV kink is present but weak on the α band. This can be seen in the extracted Eliashberg function (Fig. 2e of the main text). The 12-meV kink structure on the β band indeed becomes weaker from the Γ -K direction to the Γ -M direction. In contrast, the 32-meV kink shows opposite behavior and becomes stronger along the Γ -M direction. This non-trivial momentum- and energy-dependent electron-phonon coupling is also captured by analyzing the ratio between the band velocities below and above the kink energy, as shown in Fig. R1.3.

Generally speaking, the electron-phonon coupling vertex shows energy and momentum dependence. It can be strongly affected by the symmetry of the phonon modes, the orbital characters of the electronic bands, and the electronic density of states, *etc.* For *d*-electrons, both the orbital characters and electronic density of states show strong momentum dependence, and therefore a momentum-dependent electron-phonon coupling is expected. Interestingly, the opposite evolutions of the 12-meV and 32-meV kinks on the β -band yield a nearly momentum-independent λ .

We added this discussion in the revised supplementary materials (supplementary note 3).

Figure R1.3. Momentum evolution of the 12-meV and 32-meV kinks on the β band of CsV_3Sb_5 . (a1)-(a9), ARPES intensity plots for the cuts shown in (b). (c)-(d), Extracted band dispersions. The black lines in c are the linear fits at the energy range of $[-15, -30]$ meV while the black lines in (d) are the linear fits at the energy range of $[-9, 0]$ meV. (e) Evolution of the change ratio $(v_b/v_a - 1)$ between the band velocities below (v_b) and above (v_a) the 12-meV and 32-meV kinks.

Comment 6: 5) The momentum distribution curve at E_F shown in Fig. 1d does not look like a single peak. Was the sample free from superposition with any misaligned domain? I am aware of that CsV_3Sb_5 crystals cleave well but can leave small pieces/domain behind.

Reply 6: Since our measurements were performed in the CDW phase, we expect multiple CDW domains within our spot size ($\sim 100 \mu\text{m}$). Compared to the main peak, this side peak is relatively weak and does not affect the MDC fittings (see Fig. R1.4a). During the revision, we measured more samples, and one of them shows a single Lorentzian peak at the E_F (see Fig. R1.4b). In this “domain-free” sample, the extracted MDC width is nearly the same as samples with “domains”, confirming the robustness of our analysis.

Moreover, the momentum distribution of the electron-phonon coupling constants λ_{dev} extracted from the slope of the real-part self-energy at E_F for these two independent samples are consistent, as shown in Fig. R1.4f.

Figure R1.4. Repeat measurements for the kinks on the α band. **a**, MDCs at E_F on sample 1. **b**, Same with **a** but on sample 2. The black lines are the single-peak Lorentzian fits. **c**, Real part of the self-energies $\text{Re}\Sigma$ on sample 1 for the cuts shown in **e**. **d**, Same with **c** but for sample 2. **f**, Comparison of the EPC strength λ_{dev} measured on the samples 1 and 2.

Comment 7: 6) Note that evaluation for lambda using the slope of the self-energy may be inappropriate as temperature increases so it becomes non-negligible compared to the Debye temperature (E.W. Plummer et al., Prog. Surf. Sci. 74, 251 (2003)).

Reply 7: We fully agree with Reviewer #1. Our experiments were performed at $T = 6$ K, much smaller than the Debye temperature of CsV_3Sb_5 , ~ 198 K. Therefore, the temperature effect can be neglected for our case.

We added a note in the main text to reflect this point.

Review #2:

Thank you for your careful review of our manuscript. Below we copied your report *in black* and describe how we have edited the manuscript in response to your comments *in blue*.

Comment 1: Overall, the authors present high quality experimental data on an interesting material, however, it is a stretch to directly draw such a strong conclusion that the superconductivity originates from the EPC based on current results, because the connection between the EPC strength and the underlying superconducting pairing glue is not straightforward. Extensive works have been recently done on the superconductivity of CsV₃Sb₅ by various techniques, but few of them are referred and discussed here on the basis of current results. Rather than the brief arguments in the manuscript, the implications of the experimental data and the connections with previous studies are suggested to be further explored and discussed.

Reply 1: First, we apologize for any ambiguities delivered by the manuscript. The key message of our paper is that *electron-phonon coupling in CsV₃Sb₅ is strong enough to give rise to the superconductivity with a transition temperature comparable to the experimental value. As Reviewer #1 also pointed out: “the demand for knowledge on EPC in AV₃Sb₅ kagome metals is huge among the community”*. This is partially because the DFT calculated EPC strength λ is too small for $T_c \sim 2.6$ K [Ref. 21, *Physical Review Letters* **127**, 046401 (2021)]. Motivated by this conclusion, many theoretical and experimental works neglected BCS superconductivity in their starting point. However, as we showed in our paper, the DFT predicted λ is significantly smaller than the experimental value, and therefore, the BCS pairing mechanism cannot be excluded for CsV₃Sb₅. In our opinion, this is a very important information to correctly understand all the interesting properties of CsV₃Sb₅.

Following Reviewer #2's suggestions, in the revised manuscript, we added new comments and references to discuss the relation between BCS superconductivity and time-reversal symmetry breaking. Due to growing number of publications on this topic, we have to drop off important references that are less relevant to our study. However, if Reviewer #2 could point out key missing references, we'd be happy to cite.

Comment 2: Moreover, the analysis is not sufficient to unambiguously determine the value of EPC strength with only the real part of the self-energy being considered, it is suggested to further include the analysis via the relation between the imaginary part of the self-energy and the Eliashberg function. Hence, I don't recommend the publication in Nature Communications and suggest transferring the manuscript to a more specialized journal. Below I make some further comments.

Reply 2: Analysis of the Eliashberg function using both real- and imaginary-part self-energy is also suggested by other Reviewers. Following your suggestions, in the revised manuscript, we extracted Eliashberg function based on (1) real-part self-energy [Ref. 30, *Phys. Rev. Lett.* **92**, 186401 (2004)]; (2) imaginary-part self-energy [*Nature Communications* **5**, 3257 (2014)]; (3) self-consistent real- and imaginary-part self-energy linked through the Kramers-Kronig (K-K) relation [*Phys. Rev. B* **71**, 214513 (2005)]. All these analyses yield consistent results and establish that the

experimental electron-phonon coupling is twice larger than the DFT predicted value. Therefore, a BCS superconducting pairing cannot be excluded in CsV_3Sb_5 .

These new analyses are added in the revised supplementary materials (see supplementary note 2 and Fig. S2 or Fig. R2.2).

In the main text of the revised manuscript, we further added new experimental data on an isovalent-substituted $\text{Cs}(\text{V}_{0.93}\text{Nb}_{0.07})_3\text{Sb}_5$ with $T_c \sim 4.4$ K. Comparing with the undoped CsV_3Sb_5 , the extracted EPC strength is enhanced by about 50% on the V 3d band (Fig. 4 of the revised manuscript). Based on the extracted Eliashberg function and the enhanced electron-phonon coupling, we obtained an enhanced T_c close to the experimental value. This new result further supports EPC as a key player for the emergence of superconductivity in CsV_3Sb_5 .

Comment 3: In Fig. 1e, the correspondence is not clear by showing the FWHM of MDCs together with the real part of the self-energy. It is suggested to present the imaginary part of the self-energy instead of FWHM.

Reply 3: Thanks for the suggestion. *We added the energy scale on left axis of the Fig. 1e to indicate the imaginary part of the self-energy.*

Comment 4: In Line 74-76 (Fig. 2d), an additional kink is observed around 12 meV, which is prominent in the beta band and less prominent in the alpha band, what is the cause of this orbital dependent behavior? Can we obtain some illumination from this for the superconductivity?

Reply 4: The electron-phonon coupling vertex generally shows energy and momentum dependence. It can be strongly affected by the symmetry of the phonon modes, the orbital character of the electronic bands, the electronic density of states, *etc.* We therefore expect some orbital dependence of the electron-phonon coupling.

Reviewer #2 raised a very interesting question regarding the superconductivity. As shown in Fig. 4 of the revised manuscript, the coupling between the V 3d electrons and low energy phonon is selectively enhanced in $\text{Cs}(\text{V}_{0.93}\text{Nb}_{0.07})_3\text{Sb}_5$ with $T_c \sim 4.4$ K. This result may indicate that the coupling between 3d-electrons and phonon modes plays a more crucial role in the emergence of the superconductivity.

We added “this doping dependent enhancements of λ_d and T_c may indicate that the V 3d-electron-phonon couplings are the main driver of the superconductivity in CsV_3Sb_5 ” in line 128 of the revised main text.

Comment 5: The argument on the phonon hardening effect is confusing. The self-energy anomaly of the beta band around 12 meV remains almost the same upon warming (Fig. S4d), while the neutron studies (Ref. 26) show no evident change of the phonon energy above T_{cdw} . Why the 3d-EPC only hardens the phonons when $T < T_{\text{cdw}}$?

Reply 5: As shown in Fig. R2.1, the temperature-dependence of the first derivative of $\text{Re}\Sigma$ at E_F , $\lambda_{\text{dev}}(T)$, non-trivially shows a inflection point at the T_{CDW} , implying an enhanced EPC in the CDW

state. Since the formation of CDW enhances the phononic density-of-states, we speculate that the enhanced EPC is due to the phononic band folding. The strength of the phononic band folding is proportional to the CDW order parameter and hence expected to show temperature dependence only below the T_{CDW} .

Fig. R2.1. Inflection of the EPC induced renormalization at T_{CDW} . **a**, ARPES intensity plots of the α and β bands along Γ -K direction of CsV_3Sb_5 below and above $T_{CDW} \sim 94$ K. **b**, Extracted band dispersions at the representative temperatures. **c-d**, Temperature-dependent real part of the self-energy $\text{Re}\Sigma$ for the α and β bands, respectively. **e**, Temperature evolution of the first derivative of $\text{Re}\Sigma$ at E_F for the α and β band.

Comment 6: In Fig. S4, the kink around 30 meV in the beta band becomes much less prominent at higher temperature, in contrast to that of the alpha band, what is the cause of the different temperature dependence?

Reply 6: At low temperatures, the 32-meV and 12-meV kink in the β band along the Γ -K direction have comparable magnitudes. At higher temperatures, the dip between two peaks in the $\text{Re}\Sigma$ plot is filled due to the thermal broadening effect, making the 32-meV kink less prominent. For the α band, since the 12-meV kink is much weaker than the 32-meV kink at low temperatures, thus the 32-meV kink is still prominent at higher temperatures despite the thermal broadening.

We clarified this point in the revised supplementary materials (supplementary note 6).

Comment 7: The Eliashberg function analysis in the current manuscript is not sufficient to obtain an unambiguous EPC strength because only the real part of the self-energy is examined, which depends strongly on the choice of the bare band, quite different lambda values could be obtained by a small change of the bare band. While not doubting the current results, the authors are suggested to further examine the EPC strength by the Eliashberg function analysis with calculating both the real part and imaginary part of the self-energy.

Reply 6: We thank the referee for the constructive suggestion. To examine the influence from the choice of the bare velocity, we did a crosscheck on the extraction of the Eliashberg function $\alpha^2F(\omega)$ by a self-consistent fit procedure following: (1) we assume an initial v_0 , and calculate the $\text{Re}\Sigma$ and $\text{Im}\Sigma$ from the experimental band dispersion; (2) deduce the $\alpha^2F(\omega)$ from the $\text{Im}\Sigma$ via maximum entropy method (MEM); (3) calculate a $\text{Re}\Sigma^*$ from the $\alpha^2F(\omega)$ following the Eq. (1) in the methods of the main text. By changing the v_0 and repeating steps (1)-(3), we can eventually find a proper v_0 to minimize the difference between the $\text{Re}\Sigma$ and $\text{Re}\Sigma^*$. This self-consistent fit procedure not only extracts the $\alpha^2F(\omega)$ but also finds out the fitted bare velocity.

We performed such a self-consistent fit on the α and β bands of CsV_3Sb_5 and presented the results in Fig. R2.2c (Fit #3). As shown in Fig. 2.2d, the extracted $\alpha^2F(\omega)$ is in good consistency with the $\alpha^2F(\omega)$ extracted from the $\text{Re}\Sigma$ and $\text{Im}\Sigma$ based on the bare velocity assumed in the main text, except for a deviation around ~ 12 meV on the α band. This could be explained by that strength of the 12-meV kink is much weaker compared to the 32-meV kink on the α band thus the uncertainties of the MEM fits are increased. As shown in Fig. R2.2e, the EPC strength λ estimated from the $\alpha^2F(\omega)$ from these analyses are both located in the range of 0.45~0.6, which we used to estimate McMillan's superconducting transition temperature. More importantly, the fitted bare band velocity v_0 is quite close to the one assumed in the main text. These all validate the bare band assumption used in our main manuscript.

These results are added to the revised supplementary materials (see supplementary note 2 and Fig. S2).

Figure R2.2. Comparison of the Eliashberg function $\alpha^2 F(\omega)$ from different fitting procedures. a, Real parts of the self-energy $\text{Re}\Sigma$ and the corresponding fits to extract the $\alpha^2 F(\omega)$ (fit #1, solid red line in d). **b**, Imaginary parts of the self-energy $\text{Im}\Sigma$ and the corresponding fits to extract the $\alpha^2 F(\omega)$ (fit #2, dashed line in d). The $\text{Re}\Sigma$ and $\text{Im}\Sigma$ in **a** and **b** are extracted based on the assumed bare velocity v_0 . **c**, Self-consistently fits with the trial bare velocities (fit #3). The extracted $\alpha^2 F(\omega)$ is shown as solid black lines in **d**. **d**, Comparison of the $\alpha^2 F(\omega)$ extracted from the fits #1, #2, and #3. **e**, Estimated EPC strength λ and Debye temperature θ_D from the fits #1, #2, and #3. The α and β bands in this figure are the ones shown in Fig. 2 of the main text.

Comment 8: In Line 108-109, “We shall note that while the delta band also displays strong self-energy anomaly near the same energy scale”, this argument is confusing. Although two kink-like features are observed in the delta band in Fig. S3, the one close to E_F could be mainly affected or induced by the CDW gap opening of the delta band instead of EPC, as illustrated in Fig. S3b, S3c, and S3f, where the larger CDW gap size the flatter delta band near E_F . It is suggested to discuss the contribution of CDW gap opening in this kink-like structure.

Reply 8: We agree that the CDW gap flattens the δ band near E_F and make the EPC strength overestimated. We added “in addition, the CDW gap flatten the δ bands near E_F , hindering the estimation of the EPC strength” to point out this.

We carefully checked that the CDW gap does not open on the α and β bands, as the symmetrized EDCs shown in Fig. R2.3. Therefore, the EPC analysis for the α and β bands is reliable.

Figure R2.3. ARPES spectra symmetrized at E_F for CsV_3Sb_5 . **a**, Symmetrized ARPES intensity plots for the α bands. **b**, Symmetrized EDCs at k_F of the α bands. **c-d**, Same with **a-b** but for the β bands.

Comment 9: As implied by the arguments in Line 110-111 (main text) and Line 63-64 (supplementary), CDW seems to compete with superconductivity for the available density of states near E_F . Furthermore, as shown in Fig. S3f, the CDW gap of the delta band shows an evident momentum dependence as in Ref. 22. Therefore, one would expect an anisotropic superconducting gap opening of the delta band due to the competition between CDW and superconductivity. This could contradict the current conclusion drawn in the manuscript.

Reply 9: We are not entirely clear about this concern. We assume Reviewer #2 is questioning on an anisotropic superconducting gap versus BCS superconductivity. Theoretically, BCS superconductivity allows anisotropic gap function. There are many CDW materials, such as transition-metal dichalcogenides, that possess competing superconductivity and CDW. While the pairing mechanism is not fully determined for those materials, it is widely believed that the superconductivity is driven by electron-phonon coupling [Ref. 37, *Science* **372**, 1447-1452 (2021)].

In addition, we shall point out that the CDW order does not develop a full gap near E_F . Like the cuprate high- T_c superconductors, there is a finite spectral weight inside the CDW gap, as the symmetrized EDCs shown in Fig. R2.4. This finite spectral weight could be condensed and further open a superconducting gap below T_c . However, how the CDW gap on the δ band competes with the superconducting gap needs further measurements.

Figure R2.4. Symmetrized EDCs for the bands on the δ Fermi surface (FS). **a**, FS mapping of the CsV_3Sb_5 sample. **b**, Symmetrized EDCs at k_F of the δ bands along the cuts shown in **a** (black lines).

Comment 10: Related to the above point, although the EPC strength can support a T_c comparable to the experimental value, here the connection between the EPC strength and the underlying superconducting pairing glue in CsV_3Sb_5 is not straightforward. In order to draw such a strong conclusion, the authors are suggested to measure the superconducting gap symmetry in the momentum space. Otherwise, I suggest the authors could soften this point in the manuscript.

Reply 10: Again, we apologize for the ambiguity delivered in our previous manuscript. In the revised manuscript, (1) we clearly described our main message that *EPC in CsV_3Sb_5 is strong enough to give rise to the superconductivity with a transition temperature comparable to the experimental value*; (2) we observed enhanced EPC on an isovalent-substituted $\text{Cs}(\text{V}_{0.93}\text{Nb}_{0.07})_3\text{Sb}_5$ with $T_c \sim 4.4$ K, supporting EPC as a key player for the emergence of superconductivity in CsV_3Sb_5 .

Comment 10: 9. Some typos in the supplementary. Line 60, Fig. S3d \rightarrow Fig. S3f; Fig. S3a, gamma \rightarrow delta; Line 80, liens \rightarrow lines.

Reply 10: Thanks for the reminder. We corrected them in the revised manuscript.

Reviewer #3:

Thank you for your careful review of our manuscript. Below we copied your report *in black* and describe how we have edited the manuscript in response to your comments *in blue*.

First, we would like to summarize the main improvements in the revised manuscript.

- (1) We added new experimental data of an isovalent-substituted $\text{Cs}(\text{V}_{0.93}\text{Nb}_{0.07})_3\text{Sb}_5$ with $T_c \sim 4.4$ K. Our self-energy analysis shows that the EPC strength is enhanced by about 50% on the V 3d band (Fig. 4 of the revised manuscript). Based on the extracted Eliashberg function, we obtained an enhanced T_c close to the experimental value. This new result strengthens the main conclusion of this work that EPC plays a key role for the emergence of superconductivity in CsV_3Sb_5 .
- (2) We extracted the Eliashberg function using real- and imaginary-part self-energy and checked their consistency using the Kramers-Kronig relation.

Comment 1: The authors extract λ at low temperature. However, based on Fig. S4 where the temperature dependence of $\text{Re}\Sigma$ is presented for both alpha and beta bands, the kink strength decreases when T increases (suppression of around 20-40% for both alpha and beta bands – the weakening of the kink is easily observable by superimposing $\text{Re}\Sigma$ curves at different T). Given that (i) there is no clear dependence of $\text{Re}\Sigma$ on the CDW transition, and (ii) it is unlikely that the EPC is changing over T, the authors should extract λ at high temperature where the CDW is absent. Indeed, any gapping of the electronic states (even if localized in a few areas of the momentum space) may lead to a positive feedback effect enhancing the kink strength [for instance, see Li et al. Nature Communications 9, 26 (2018)].

Reply 1: We thank Reviewer #3's constructive suggestion. We would like to point out that there is a clear change of $\text{Re}\Sigma$ at the CDW transition temperature (T_{CDW}). As shown in Fig. R3.1, the first derivative of $\text{Re}\Sigma$ at E_F , λ_{dev} , shows an inflection at T_{CDW} and becomes larger below T_{CDW} , implying an enhanced EPC in the CDW state.

We believe the meaning of the “positive feedback effect” mentioned by Reviewer #3 is twofold. First, the formation of CDW can intrinsically enhance the EPC due to the change of the electronic and phononic density-of-states. Second, the CDW gap will change the electronic structure near E_F , therefore the linear bare band assumption is no longer valid (because of the band bending effect). In the latter case, the EPC strength will be extrinsically enhanced if the bare band is not properly chosen. In CsV_3Sb_5 , as we show in Fig. R3.2, for the α and β bands the linear bare band assumption works well at $T = 6$ K and gives a self-consistent single-particle self-energy (also see the supplementary note 2 and Fig. S2). Therefore, the observed CDW-EPC feedback effect is intrinsically induced by the change of both the electronic and phononic density-of-states due to the formation of the CDW order. As the phase diagram shown in Fig. 1a of the main text, superconductivity shows a double dome structure and is intimately correlated with the CDW via doping and pressure [Refs. 33-36], suggesting that the CDW-EPC feedback effect is likely critical for the superconductivity in CsV_3Sb_5 . Therefore, we think evaluating EPC in the CDW phase is more relevant to superconductivity.

Besides, due to the Fermi-Dirac distribution is the electron-phonon scattering vertex, the derivative approach is exact only at $T=0$ and approximately accurate at $T \ll \theta_D$, where $\theta_D \sim 198$ K is the Debye temperature in CsV_3Sb_5 . Because of these considerations, we believe it is better to extract the EPC strength just above T_c .

Fig. R3.1. Inflection of the EPC induced renormalization at T_{CDW} . **a**, ARPES intensity plots of the α and β bands along Γ -K direction of CsV_3Sb_5 below and above $T_{CDW} \sim 94$ K. **b**, Extracted band dispersions at the representative temperatures. **c-d**, Temperature-dependent real part of the self-energy $\text{Re}\Sigma$ for the α and β bands, respectively. **e**, Temperature evolution of the first derivative of $\text{Re}\Sigma$ at E_F for the α and β band.

Comment 2: The only way to extract a reliable λ is to analyze the kink for $T > T_{CDW}$. However, I am afraid that for $T > T_{CDW}$ the extracted λ will be smaller than what is reported in the current manuscript (and possibly matching with DFT calculations), thus invalidating the authors' conclusions.

Reply 2: As we replied above, we think the λ in the CDW phase is more relevant to superconductivity in CsV_3Sb_5 .

Comment 3: The authors extract the Eliashberg function and related λ using the maximum entropy method (MEM), as shown in Fig. 2. In this regard, what bare dispersion was used? In addition to the $\text{Re}\Sigma$ of Fig. 2d, the authors should display the $\text{Im}\Sigma$ extracted via MEM and compare it to the experimental $\text{Im}\Sigma$.

Reply 3: This is a very good point and critical for our manuscript. We used a linear dispersion connecting $E_B = 100$ and 0 meV as the bare band to extract the self-energy. As shown in Fig. R3.2, the Kramers-Kronig transformation of the imaginary-part self-energy overlaps well with the real-part self-energy. The good consistency validates the bare-band assumption used in our manuscript. *We added this comparison in the revised supplementary materials (supplementary note 1 and Fig. S1).*

Following Reviewer #3's suggestion, we compared the Eliashberg functions extracted from the real- and imaginary-part self-energy via MEM. As shown in *Fig. S2 in the revised manuscript*, they have a good consistency.

Finally, as a cross-check, we employed the self-consistent fit of both real- and imaginary-part self-energy starting from a trial bare velocity [*Phys. Rev. B* **71**, 214513 (2005)]. The converged bare band velocity validates our bare-band assumptions used in other methods (*supplementary note 2 and Fig. S2 in the revised manuscript*).

Figure R3.2. Consistency-check of the self-energy following Kramers-Kronig relation. **a**, Imaginary part of the self-energy $\text{Im}\Sigma$ (top panel) and real part of the self-energy $\text{Re}\Sigma$ (bottom panel) for the α band. **b**, Same with **a** but for the β band. The black lines in the bottom panel are the real parts transformed from $\text{Im}\Sigma$ following the Kramers-Kronig relation. A constant background is subtracted for $\text{Im}\Sigma$. The α and β band here are the ones presented in Fig. 2 of the main text.

Comment 4: Moreover, the authors estimate the momentum dependence of λ by computing the first derivative of $\text{Re}\Sigma$ at E_F . This method is unreliable for quantitative extraction of λ (it is blind to other possible underlying contributions, such as e-e). Indeed, the λ extracted by MEM is approximately 50% lower than the one retrieved by the first derivative of $\text{Re}\Sigma$, clearly showing the unreliability of the first-derivative method. The authors should apply the MEM at high temperature to extract the momentum-dependence of λ .

Reply 4: We would kindly point out that (1) the derivative method is precise for $T = 0$ K and valid for $T \ll \theta_D$ [Plummer et al., *Prog. Surf. Sci.* **74**, 251 (2003) and Ref. 31]. Since our analysis was based on $T = 6$ K data ($\ll \theta_D \sim 198$ K), the derivative method is a good approximation. (2) The MEM method is sensitive to the experimental errors in $\text{Re}\Sigma$ and $\text{Im}\Sigma$ and the initial guess of the Eliashberg function. Mathematically, it is difficult to control these errors. The key point of our analysis is that the experimentally extracted electron-phonon coupling is much larger than the DFT predicted value and therefore a BCS superconducting pairing mechanism cannot be excluded for CsV_3Sb_5 .

Reviewers' Comments:

Reviewer #1:

Remarks to the Author:

I see that the authors provide answers to raised questions by referees and improved their manuscript following the comments and suggestions. I still think that the Information provided by this paper such as detailed momentum-dependent electron-phonon coupling strength is valuable. Meanwhile, it should be noted that this work is just demonstrating characteristics that fits well with BCS, and a full conclusion saying "key role of EPC on the superconductivity" is too hard. If authors' tentative future work with examinations on Nb-doped variants, then that future work may contain such an argument but the current one which only has a "comparison". When going through the revised manuscript, I found more strong arguments, so the current paper still needs to soften its conclusion as other referees pointed out. I also have some questions for technical details. Also, the authors may need to put more kind explanations for general readers. Here are points I would like authors to address.

- 1) In the abstract, "establish the key role of EPC on the superconductivity" (line 31-32) is too strong as this work did not navigate EPC in the phase diagram, for example, detailed Nb doping dependence of EPC.
- 2) In the same context, regarding the line 51 in the main text, I do not think this work provides "doping dependence".
- 3) In the line 49, the authors wrote "experimental estimation of lambda and its relevance with superconductivity in CsV3Sb5 are still missing". Actually, the reference 22 estimated the lambda from the delta band (meanwhile I agree with authors that because of CDW gap it is not reliable to infer the EPC strength from the delta band). Please cite their work in a proper way and soften the sentence.
- 4) The introduction part may contain too general information about kagome metals and has me distracted from the main conclusion of this work. For instance, are references 6-8 really relevant to this work?
- 5) I elaborated in the previous report that the momentum distribution curve shown in Fig. 1d is not a single peak. The authors' response seems reasonable in their letter, while they still contain the same figure in the main text without revision. In Fig. 1d, the black fitting curve is obviously not single Lorentzian. Please explain it in detail somewhere.
- 6) In the line 81, how can the conclusion "the giant phonon hardening is induced primarily by the V 3d-orbitals" be drawn? Can they distinguish causes/consequences from their results?
- 7) I am satisfied that the authors now provide more comprehensive analysis results based on Kramers-Kronig (K-K) relation and so on. Overall, it looks fine, but in Fig. 1e, the label for the imaginary part of the self-energy (ImSE) looks weird as it becomes zero at EF while FWHM does not.
- 8) Another note regarding the ImSE is that ImSE needs to be fitted with e-e interactions and e-ph modes to point out "energy scale" of the EPC as provided by ARPES works for other systems [for an example, T. L. Yu et al., Nat. Commun. 13, 6560 (2022)].
- 9) Another point regarding K-K; in the supplementary materials, the authors may explain the K-K method for general readership. In the current version, there is even no reference cited in that section.
- 10) In the line 84 in the supplementary materials, the authors describe about the Fig. S3, not Fig. S4. Please correct it.

Reviewer #2:

Remarks to the Author:

The authors have done a mainly good job of responding to my earlier points. Before I can change my recommendation to support the publication of this paper, the authors should discuss more on the CDW gap of delta band and its controversial relationship with superconductivity in the manuscript. The current writing leads to think that the SC gap could be anisotropic since its competition with the anisotropic CDW gap. Indeed, the BCS superconductivity allows anisotropic gap function, but I have the impression the anisotropy contradicts the other paper on SC gap measurements from the same

authors.

Reviewer #3:

Remarks to the Author:

I acknowledge that the authors did an excellent job answering the criticisms raised during the first round of review. Moreover, the self-consistent KK analysis, as well as the addition of the analysis of Cs(V_{0.93}Nb_{0.07})₃Sb₅ (Fig. 4), have definitely improved this manuscript.

However, I am still not convinced that the extracted EPC values may be quantitatively reliable. In reply 8 to reviewer #2, the authors write: "We agree that the CDW gap flattens the d band near EF and makes the EPC strength overestimated. We added in addition, the CDW gap flatten the d bands near EF, hindering the estimation of the EPC strength to point out this. We carefully checked that the CDW gap does not open on the a and b bands, as the symmetrized EDCs shown in Fig. R2.3. Therefore, the EPC analysis for the a and b bands is reliable."

The fact that the CDW gap does not open on the α and β bands does not preclude that it may still affect the EPC extracted for those bands. Indeed, Fig. 3f clearly highlights that the kink strength is isotropic in momentum for the α and β bands. This evidence points towards the contribution of isotropic phonon modes (similar to the 80 meV bond stretching mode in cuprates) connecting different states over the whole momentum space. Therefore, although α and β bands do not present any CDW gap, the CDW gap at δ might still affect the EPC extraction for those bands (the phonon's q-vector may connect the α and β bands to the gaped δ band). For this reason, I previously recommended that the authors extract λ for $T > T_{CDW}$. I understand that superconductivity emerges within the CDW phase wherein EPC could be enhanced, however a similar enhancement of the real part of the Self-Energy could originate from the appearance of electronic gaps.

In addition, one should note that a quantitative extraction of λ via the first derivate of $\text{Re}\Sigma$ requires a non-gapped electronic DOS and low-energy phonon modes (i.e., initial and final scattering states close in energy), conditions that do not hold in the present case. λ extracted via the first derivative of $\text{Re}\Sigma$ is a qualitative estimate.

Overall, although I still have some doubts as discussed above, I do believe that Nature Communications could be a good venue to publish these high-quality ARPES results. However, before proceeding, I recommend the authors mention all possible contributions that might affect any quantitative extraction of λ , thus partially softening their claims.

Reviewer #1:

We thank you for your careful review of our manuscript. Below we copied your report *in black* and describe how we have edited the manuscript in response to your comments *in blue*.

I see that the authors provide answers to raised questions by referees and improved their manuscript following the comments and suggestions. I still think that the Information provided by this paper such as detailed momentum-dependent electron-phonon coupling strength is valuable. Meanwhile, it should be noted that this work is just demonstrating characteristics that fits well with BCS, and a full conclusion saying “key role of EPC on the superconductivity” is too hard. If authors’ tentative future work with examinations on Nb-doped variants, then that future work may contain such an argument but the current one which only has a “comparison”. When going through the revised manuscript, I found more strong arguments, so the current paper still needs to soften its conclusion as other referees pointed out. I also have some questions for technical details. Also, the authors may need to put more kind explanations for general readers. Here are points I would like authors to address.

Comment 1: In the abstract, “establish the key role of EPC on the superconductivity” (line 31-32) is too strong as this work did not navigate EPC in the phase diagram, for example, detailed Nb doping dependence of EPC.

Reply 1: Following Reviewer #1’s suggestion, we softened our conclusion in the revised manuscript. We changed the last sentence of the abstract to be “*our results provide an important clue to understand the pairing mechanism in the kagome superconductor CsV₃Sb₅*” and correspondingly modified the conclusion in the main text.

Comment 2: In the same context, regarding the line 51 in the main text, I do not think this work provides “doping dependence”.

Reply 2: In the revised manuscript, we removed the expression of “*doping dependence*”.

Comment 3: In the line 49, the authors wrote “experimental estimation of lambda and its relevance with superconductivity in CsV₃Sb₅ are still missing”. Actually, the reference 22 estimated the lambda from the delta band (meanwhile I agree with authors that because of CDW gap it is not reliable to infer the EPC strength from the delta band). Please cite their work in a proper way and soften the sentence.

Reply 3: We would like to thank Reviewer #1 for the important notification. The reference *Nature communications* **13**, 273 (2022) (Ref. 22) made the first experimental report of electronic kink on δ band. They, however, didn’t extract the EPC strength λ , which is one of the main focuses of the work.

In the revised manuscript, we modified this sentence to be “However, recent ARPES study of a cousin compound KV₃Sb₅ revealed a clear electronic kink²² in the electronic structure near the van Hove singularity, suggesting a moderate EPC. Therefore, an experimental estimation of

orbital- and momentum-dependent λ and its possible connection with superconductivity are highly desired to understand the nature of the superconductivity in CsV_3Sb_5 .”

Comment 4: The introduction part may contain too general information about kagome metals and has me distracted from the main conclusion of this work. For instance, are references 6-8 really relevant to this work?

Reply 4: We would like to thank Reviewer #1 for the constructive suggestion. We modified the introduction part by focusing on the debate on the superconducting mechanism in CsV_3Sb_5 (see the revised manuscript).

Comment 5: I elaborated in the previous report that the momentum distribution curve shown in Fig. 1d is not a single peak. The authors’ response seems reasonable in their letter, while they still contain the same figure in the main text without revision. In Fig. 1d, the black fitting curve is obviously not single Lorentzian. Please explain it in detail somewhere.

Reply 5: We mistakenly forgot to change the fitted curve shown in Fig. 1d in the last round and we apologize for this mistake. The MDC at E_F shown in previous Fig. 1d was fitted using a double-peak Lorentzian function. As we presented in the previous response letter, this side-peak is too weak to affect the accuracy of the single-peak Lorentzian fits, and our results can be repeated from an independent experiment performed in a “domain-free” sample.

In the revised manuscript, we plotted ARPES data on a “domain-free” sample and added the corresponding Fermi surface mapping in Fig. 1c. In the revised supplementary materials (supplementary note 5), we added a discussion on the effect of self-energy analysis.

Comment 6: In the line 81, how can the conclusion “the giant phonon hardening is induced primarily by the V 3d-orbitals” be drawn? Can they distinguish causes/consequences from their results?

Reply 6: We reached this conclusion based on the following experimental observations: (1) X-ray, neutron and optical studies have shown a phonon hardening effect at 12 meV below T_{CDW} [Refs. 8 and 26-28 in the previous manuscript]; (2) A 12-meV electronic kink is clearly observed in the CDW phase; (3) The extracted $\text{Re}\Sigma(\omega)$ shown in Fig. 2d and Eliashberg function shown in Fig. 2e consistently show a stronger electron-phonon coupling strength on the β band, which primarily has V 3d-orbital characters; (4) The temperature dependence of electron-phonon coupling on V 3d-bands at $\omega \sim 12$ meV show an enhancement below T_{CDW} , consistent with the X-ray, neutron and optical studies. We therefore concluded that “the giant phonon hardening is induced primarily by the V 3d-orbitals”.

In our perspective, the stronger 12-meV kink on the β band and the phonon hardening at 12 meV are both consequences of the V 3d-electron phonon coupling. However, we should admit that our observation couldn’t rule out other mechanisms for the giant phonon hardening effect, such as a CDW amplitude mode coupling with optical phonon modes near 12 meV.

To avoid potential ambiguities, we removed this sentence in the revised manuscript.

Comment 7: I am satisfied that the authors now provide more comprehensive analysis results based on Kramers-Kronig (K-K) relation and so on. Overall, it looks fine, but in Fig. 1e, the label for the imaginary part of the self-energy (ImSE) looks weird as it becomes zero at EF while FWHM does not.

Reply 7: For a Fermi liquid ground state, $\text{Im}\Sigma(\omega=E-E_F=0) = 0$ at $T = 0$, corresponding to a delta function for the MDC at E_F . However, due to finite instrument resolution and the binding-energy independent electron-impurity scatterings, the MDC width at E_F is always finite. Therefore, practically [Ref. 23, *Phys. Rev. Lett.* **83**, 2085 (1999)] a constant background will be subtracted from $\text{Im}\Sigma(\omega)$ after converting MDC width to $\text{Im}\Sigma(\omega)$. As we noted in the revised supplementary materials, the K-K transformation of a constant C_0 is $P \int_{-\infty}^{+\infty} C_0/(\omega' - \omega)d\omega' = 0$. Therefore, removing a constant background from $\text{Im}\Sigma(\omega)$ would not change the results.

Comment 8: Another note regarding the ImSE is that ImSE needs to be fitted with e-e interactions and e-ph modes to point out “energy scale” of the EPC as provided by ARPES works for other systems [for an example, T. L. Yu et al., *Nat. Commun.* 13, 6560 (2022)].

Reply 8: We fully agree with Review #1 on this important remark. Assuming the microscopic scattering processes are independent, the main imaginary-part self-energy of a non-magnetic metal is given by $\text{Im}\Sigma = \text{Im}\Sigma_{ep} + \text{Im}\Sigma_{ele} + \text{Im}\Sigma_{imp}$ [*Phys. Rev. Lett.* **83**, 2085 (1999), *Nat. Commun.* 13, 6560 (2022)]. Here $\text{Im}\Sigma_{ep}$ is contributed from the electron-phonon couplings, which has a characteristic step feature at the energy of the coupled phonon mode. For binding energy (E_B) much higher than this step energy, $\text{Im}\Sigma_{ep}$ is nearly constant. For the electron-electron correlation part, $\text{Im}\Sigma_{ele}$ is proportional to ω^α (e.g. $\alpha = 2$ for a Fermi liquid and $\alpha = 1$ for a marginal Fermi liquid, etc.). Besides these main energy-dependent self-energy parts, electron-impurity scatterings will give rise to energy-independent self-energy $\text{Im}\Sigma_{imp}$.

Following Reviewer #1’s suggestions, we extracted $\text{Im}\Sigma_{ep}$ by assuming $\text{Im}\Sigma_{others} = \text{Im}\Sigma_{ele} + \text{Im}\Sigma_{imp} = C_0 + C_1\omega^2$. Based on the discussion above, for $\omega = E_B > 40$ meV (well beyond the kinks), the energy-dependent self-energy component is contributed solely to $C_1\omega^2$. We extracted C_1 from a parabolic fit to $\text{Im}\Sigma(\omega > 40$ meV) and C_0 was extracted from the value of $\text{Im}\Sigma$ at E_F , i.e. $C_0 = \text{Im}\Sigma_0 = \text{Im}\Sigma(\omega = 0)$. Figure R1 shows the extracted $\text{Im}\Sigma_{others}$ (purple lines) and $\text{Im}\Sigma_{ep}$ (red markers) for the α and β bands. We found $\text{Im}\Sigma_{ele}$, as indicated by the yellow shadows in Fig. R1, has a minor contribution to the total $\text{Im}\Sigma$. Quantitatively, for $E_B < 60$ meV, $\text{Im}\Sigma_{ele}$ is smaller than ~ 2 meV. This is expected because the total $\text{Im}\Sigma$ in E_B beyond the kinks is nearly linear. Since we have $\text{Im}\Sigma_{others} \simeq \text{Im}\Sigma_0 = C_0$ with $E_B < 60$ meV, the extractions of the Eliashberg functions from $\text{Im}\Sigma' = \text{Im}\Sigma - \text{Im}\Sigma_0$ via maximum entropy method still are well justified.

We added the above discussion in the revised supplementary materials (supplementary note 2) and cited the reference T. L. Yu et al., Nat. Commun. 13, 6560 (2022) (Ref. 26) in the revised main text.

Fig. R1. Subtraction of the background of $\text{Im}\Sigma_{\text{others}}$ in the width-derived raw $\text{Im}\Sigma$.

Comment 9: Another point regarding K-K; in the supplementary materials, the authors may explain the K-K method for general readership. In the current version, there is even no reference cited in that section.

Reply 9: In the revised supplementary materials, we presented the mathematical formula of the K-K transformation. We also added new references for the K-K transformation of the ARPES data.

Comment 10: In the line 84 in the supplementary materials, the authors describe about the Fig. S3, not Fig. S4. Please correct it.

Reply 10: We apologize for the mistake and thank the referee to point out this. We corrected this mistake.

Reviewer #2:

The authors have done a mainly good job of responding to my earlier points. Before I can change my recommendation to support the publication of this paper, the authors should discuss more on the CDW gap of delta band and its controversial relationship with superconductivity in the manuscript. The current writing leads to think that the SC gap could be anisotropic since its competition with the anisotropic CDW gap. Indeed, the BCS superconductivity allows anisotropic gap function, but I have the impression the anisotropy contradicts the other paper on SC gap measurements from the same authors.

Reply: This is, indeed, an excellent point. While we feel our data is hard to conclusively address this point, we would like to share our insight on this interesting remark. First, isotropic (or small anisotropic) superconductivity has been observed in other CDW materials, such as transition metal dichalcogenides [*Science* **372**, 1447-1452 (2021)]. On the other hand, we agree with Reviewer #2 that superconductivity and CDW are competing orders and hence expected to yield an anisotropic gap function, at least in the simple single-band picture. We attributed these puzzling results to the multi-orbital degrees of freedom. As shown in previous DFT calculations of AV_3Sb_5 , there are multiple van Hove singularities at M point near E_F [Ref. 21, *Phys. Rev. Lett.* **127**, 046401 (2021)]. Depending on the types and binding energies of these van Hove singularities, the orbital dependent CDW gaps are very different. Actually, even the center energies of the CDW gap can be away from the E_F . Therefore, the electronic density of state at E_F is not expected to be a clean gap as confirmed by the ARPES measurements [Ref. 20, *Nature Physics* **18**, 301–308 (2022) and Fig. S6f]. In this case, the superconducting gap near the M-point can be induced by the pair-hopping term. Consequently, a nearly isotropic SC gap function is realized.

Since these discussions are beyond the scope of the current manuscript, we added the pertinent discussions in the revised supplementary materials (supplementary note 7). In the revised main text, to avoid possible misunderstandings, we modified the related sentences to be “We shall note that the CDW gap near the M point¹⁷ (supplementary Fig. S6f) flattens the δ bands near E_F , hindering the precise estimation of EPC strength. However, the strong self-energy anomalies are observed on the δ bands and they have the energy scales same as the α and β bands (supplementary Fig. S6).”

Reviewer #3:

I acknowledge that the authors did an excellent job answering the criticisms raised during the first round of review. Moreover, the self-consistent KK analysis, as well as the addition of the analysis of Cs(V0.93Nb0.07)3Sb5 (Fig. 4), have definitely improved this manuscript.

However, I am still not convinced that the extracted EPC values may be quantitatively reliable. In reply 8 to reviewer #2, the authors write: “We agree that the CDW gap flattens the d band near EF and makes the EPC strength overestimated. We added in addition, the CDW gap flatten the d bands near EF, hindering the estimation of the EPC strength to point out this. We carefully checked that the CDW gap does not open on the a and b bands, as the symmetrized EDCs shown in Fig. R2.3. Therefore, the EPC analysis for the a and b bands is reliable.”

The fact that the CDW gap does not open on the α and β bands does not preclude that it may still affect the EPC extracted for those bands. Indeed, Fig. 3f clearly highlights that the kink strength is isotropic in momentum for the α and β bands. This evidence points towards the contribution of isotropic phonon modes (similar to the 80 meV bond stretching mode in cuprates) connecting different states over the whole momentum space. Therefore, although α and β bands do not present any CDW gap, the CDW gap at δ might still affect the EPC extraction for those bands (the phonon's q-vector may connect the α and β bands to the gaped δ band). For this reason, I previously recommended that the authors extract λ for $T > T_{CDW}$. I understand that superconductivity emerges within the CDW phase wherein EPC could be enhanced, however a similar enhancement of the real part of the Self-Energy could originate from the appearance of electronic gaps.

In addition, one should note that a quantitative extraction of λ via the first derivate of $\text{Re}\Sigma$ requires a non-gapped electronic DOS and low-energy phonon modes (i.e., initial and final scattering states close in energy), conditions that do not hold in the present case. λ extracted via the first derivative of $\text{Re}\Sigma$ is a qualitative estimate.

Overall, although I still have some doubts as discussed above, I do believe that Nature Communications could be a good venue to publish these high-quality ARPES results. However, before proceeding, I recommend the authors mention all possible contributions that might affect any quantitative extraction of λ , thus partially softening their claims.

Reply: We thank Reviewer #3 for his/her insightful suggestions. Following these suggestions, we added a new paragraph in the main text to discuss the influences of CDW order on the quantitative extraction of λ .

In the revised manuscript, we added “*Finally, we discuss the influences of CDW order on the quantitative extraction of λ at $T < T_{CDW}$. The formation of a CDW gap will modify the bare band to deviate from a linear dispersion near E_F . As we show in the supplementary Fig. S5, within the experimental resolution, we do not observe a CDW gap on the α and β bands. Therefore, for the α and β bands, the CDW modified bare band dispersion below T_{CDW} is $\sqrt{\varepsilon_0^2(k) + \Delta_{CDW}^2} \cong \varepsilon_0(k)$, where $\varepsilon_0(k) = v_0 \hbar k$ is the linear bare band dispersion above T_{CDW} . In this case, the linear bare band assumption used in our study is a good approximation. Indeed, the excellent agreement of*

Re $\Sigma(\omega)$ and Im $\Sigma(\omega)$ linked by Kramers-Kronig transformation^{23,26} validates the linear bare band assumption for the α and β bands (supplementary Figs. S1c-d). The linear bare band assumption, however, does not apply to the δ band that forms a CDW gap comparable to the kink energy^{20,31}. We also note that the formation of CDW will also modify the electronic self-energy. As we show in the supplementary Fig. S7e, λ_{dev} shows an inflection point at T_{CDW} , which may suggest an enhanced EPC strength below T_{CDW} . However, it can also be a consequence of the CDW corrected electronic self-energy effect (see supplementary note 8)."

Reviewers' Comments:

Reviewer #1:

Remarks to the Author:

The authors have improved their manuscript through two rounds of the reviews. I think that the current version of the manuscript contains reasonable level of arguments with excellent quality of ARPES dataset. Meanwhile there are still intrinsic limitations that prevents deep exploration for potential mechanism of superconductivity (i.e., CDW gap in the delta band) and some puzzles (i.e., the preprint by the same team reports an isotropic superconductivity gap function, and while here we see "accidentally" isotropic EPC and anisotropic CDW gaps), I believe that those can be addressed by following studies. Now the manuscript has my endorsement for publication. Just one minor thing to note is that in the line 23 of the supplementary materials, "phonon" should be a typo.

Reviewer #2:

Remarks to the Author:

In principle, I can recommend the publication of the manuscript in Nature Communications. But I still have one more question regarding the CDW gap. The authors wrote "Actually, even the center energies of the CDW gap can be away from the EF", I didn't find the evidence supporting this point from their current data, can the authors provide further data for this, or is there any signature reported in previous literature of AV3Sb5?

Reviewer #3:

Remarks to the Author:

The authors have replied to my questions and softened some of their claims in a satisfying manner. Therefore, I am happy to support the publication of the current manuscript.

We thank all reviewers for their efforts on reviewing our manuscript. We copied their comments *in black* and replied to these comments *in blue*.

Reviewer #1:

The authors have improved their manuscript through two rounds of the reviews. I think that the current version of the manuscript contains reasonable level of arguments with excellent quality of ARPES dataset. Meanwhile there are still intrinsic limitations that prevents deep exploration for potential mechanism of superconductivity (i.e., CDW gap in the delta band) and some puzzles (i.e., the preprint by the same team reports an isotropic superconductivity gap function, and while here we see “accidentally” isotropic EPC and anisotropic CDW gaps), I believe that those can be addressed by following studies. Now the manuscript has my endorsement for publication. Just one minor thing to note is that in the line 23 of the supplementary materials, “phonon” should be a typo.

Reply: We thank Reviewer #1 for supporting our manuscript to publish in Nature Communications. We appreciate Reviewer #1’s constructive suggestions, which will motivate us for future studies on this interesting material.

We have corrected the typo in line 23 of the supplementary materials.

Reviewer #2:

In principle, I can recommend the publication of the manuscript in Nature Communications. But I still have one more question regarding the CDW gap. The authors wrote “Actually, even the center energies of the CDW gap can be away from the E_F ”, I didn’t find the evidence supporting this point from their current data, can the authors provide further data for this, or is there any signature reported in previous literature of AV3Sb5?

Reply: We thank reviewer #2 for supporting our manuscript to publish in Nature Communications. We find that our original statement “*Actually, even the center energies of the CDW gap can be away from the E_F* ” is ambiguous. This statement is intended to emphasize the general fact that the CDW phase doesn’t have the particle-hole symmetry and hence the center of the CDW gap is not pinned at the E_F . (As a comparison, the superconductivity has the particle-hole symmetry, therefore the center of superconducting gap is pinned at E_F .) Focusing on CsV₃Sb₅, our ARPES measurements showed that the band gap in the occupied states is approximately 10 meV, however, the total CDW gap determined by the optical spectroscopy is about 86 meV [*Physical Review B*, **104**, L041101 (2021)]. These observations suggest that the center of the CDW gap is ~30 meV away from E_F .

To avoid possible misunderstandings, we have rephrased the sentence in our revised supplementary materials to be “*Actually, due to the absence of particle-hole symmetry of the CDW phase, even the center energy of the CDW gap can be away from the E_F .*”

Reviewer #3:

The authors have replied to my questions and softened some of their claims in a satisfying manner. Therefore, I am happy to support the publication of the current manuscript.

Reply: We thank the reviewer #3 for supporting the publication of our manuscript.